# NOISE-FREE SCORE DISTILLATION

**Oren Katzir**[1]*  **Or Patashnik**[1]*  **Daniel Cohen-Or**[1]  **Dani Lischinski**[2]

[1]*Tel-Aviv University*  [2]*The Hebrew University of Jerusalem*

## ABSTRACT

Score Distillation Sampling (SDS) has emerged as the de facto approach for text-to-content generation in non-image domains. In this paper, we reexamine the SDS process and introduce a straightforward interpretation that demystifies the necessity for large Classifier-Free Guidance (CFG) scales, rooted in the distillation of an undesired noise term. Building upon our interpretation, we propose a novel Noise-Free Score Distillation (NFSD) process, which requires minimal modifications to the original SDS framework. Through this streamlined design, we achieve more effective distillation of pre-trained text-to-image diffusion models while using a nominal CFG scale. This strategic choice allows us to prevent the over-smoothing of results, ensuring that the generated data is both realistic and complies with the desired prompt. To demonstrate the efficacy of NFSD, we provide qualitative examples that compare NFSD and SDS, as well as several other methods.

## 1 INTRODUCTION

Image synthesis has recently witnessed significant progress in terms of image quality and diversity (Yu et al., 2022; Ding et al., 2021; Gafni et al., 2022; Chang et al., 2022; Kang et al., 2023; Chang et al., 2023). Specifically, text-to-image models are rapidly improving, with diffusion-based methods leading the way (Sohl-Dickstein et al., 2015; Ho et al., 2020; Dhariwal & Nichol, 2021; Rombach et al., 2022; Balaji et al., 2022; Saharia et al., 2022). Seeking to project the great power of such diffusion-based text-to-image models to other domains beyond images, Score Distillation Sampling (SDS) was introduced. In their seminal work, DreamFusion, Poole et al. (2022) introduce the SDS loss which utilizes the strong prior learned by a text-to-image diffusion model to optimize a NeRF (Mildenhall et al., 2020) based on a single text prompt. Other works have shown that this mechanism can also be used to optimize other representations, such as meshes (Chen et al., 2023), texture maps (Metzer et al., 2022; Tsalicoglou et al., 2023), fonts (Iluz et al., 2023; Tanveer et al., 2023), and SVG (Jain et al., 2023).

Despite the widespread adoption of the SDS loss in various domains and representations, there is still a gap in visual quality between images generated by the standard denoising diffusion process (ancestral sampling) and those resulting from an SDS-based optimization process. Specifically, as noted by previous works (Poole et al., 2022; Wang et al., 2023b; Zhu & Zhuang, 2023), SDS tends to produce over-smoothed and over-saturated results, exhibiting limited ability to generate fine details, a trait where modern text-to-image models typically excel. Furthermore, the SDS loss remains intriguing, as it is still not fully understood.

In this paper, inspired by SDS (Poole et al., 2022), we present a general framework that allows using a pretrained diffusion model to optimize a differentiable image renderer. Treating the diffusion model as a score function (Song et al., 2020), we propose a formulation that decomposes the score into three intuitively interpretable components: alignment with the condition, domain correction, and denoising. Based on insights gained by viewing the score function in light of our new decomposition, we introduce a new Noise-Free Score Distillation (NFSD) loss, and show that it outperforms SDS without incurring any additional computational costs.

To demonstrate the general nature of our framework, we show that our novel formulation supports and provides a more concise and straightforward explanation for recent methods, such as VSD (Wang et al., 2023b) and DDS (Hertz et al., 2023), which have shown improvements over SDS.

---

* Denotes Equal Contribution

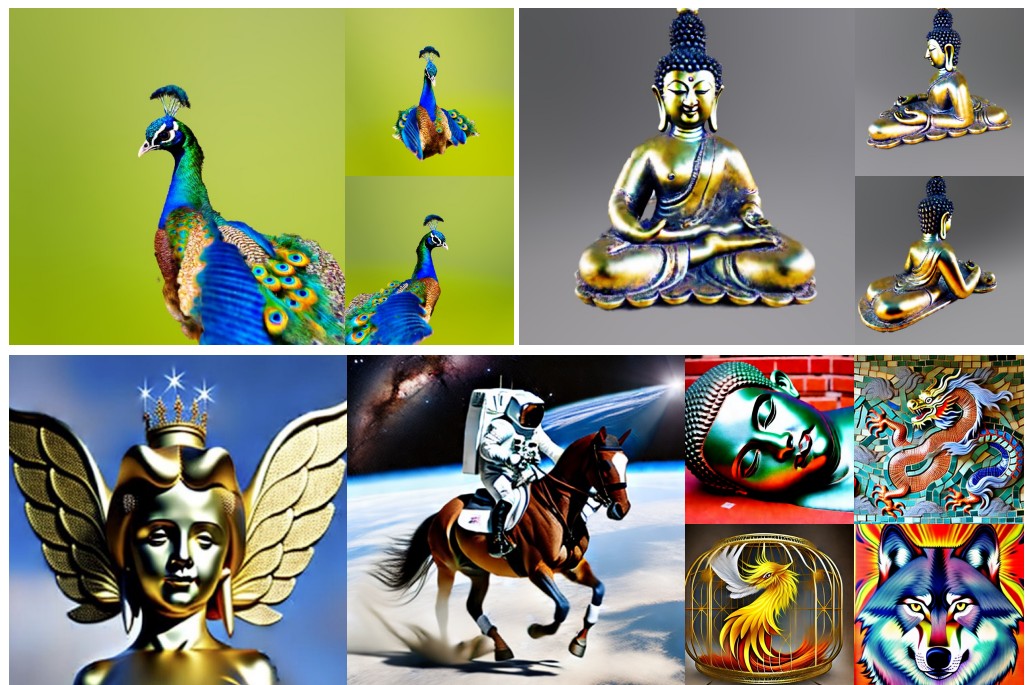

Figure 1: Results obtained with our Noise Free Score Distillation (NFSD). Top: two learnt NeRFs (movies of these and many other examples are included in the supplementary materials). Bottom: a gallery of images optimized with NFSD.

We validate our formulation and approach by utilizing Stable Diffusion (Rombach et al., 2022) as our score function with a focus on images and NeRFs as our representations. A few example results are showcased in Figure 1. Through careful design, our Noise-Free Score Distillation (NFSD) addresses some of the issues present in SDS and leads to improved visual results.

## 2 BACKGROUND

In this section, we provide the necessary background regarding diffusion models and the SDS loss (Poole et al., 2022) that enables text-to-3D generation by optimizing the parameters of a differentiable image generation function.

**Diffusion models** Diffusion models (Sohl-Dickstein et al., 2015; Ho et al., 2020) are a family of generative models that are trained to gradually transform Gaussian noise into samples from a target distribution $p_{\text{data}}$. Starting from an initial noise $\mathbf{z}_T \sim \mathcal{N}(0, \mathrm{I})$, at each diffusion timestep $t$, the model takes as input a noisy sample $\mathbf{z}_t$, and predicts a cleaner sample $\mathbf{z}_{t-1}$, until finally obtaining $\mathbf{z}_0 = \mathbf{x} \sim p_{\text{data}}$. Thus, such models effectively learn the transitions $p(\mathbf{z}_{t-1}|\mathbf{z}_t)$.

Commonly, diffusion models are parameterized by a U-net $\epsilon_\phi(\mathbf{z}_t, t)$ (Ho et al., 2020), which predicts the noise $\epsilon$ that was used to produce $\mathbf{z}_t$ from $\mathbf{x} = \mathbf{z}_0$, rather than predicting $\mathbf{x}$ or $\mathbf{z}_{t-1}$ directly. This is known as $\epsilon$-prediction. Previous works (Ho et al., 2020; Song et al., 2020) have also observed that $\epsilon_\phi(\mathbf{z}_t, t)$ is proportional to the predicted score function (Hyvärinen & Dayan, 2005) of the smoothed denisty $\nabla_{\mathbf{z}_t} \log p_t(\mathbf{z}_t)$, where $p_t$ is the marginal distribution of the samples noised to time $t$. The score function is a vector field that points towards higher density of data at a given noise level. Thus, intuitively, taking steps in the direction of the score function gradually moves the sample towards the data distribution.

In this work, we focus on diffusion models that strive to generate samples aligned with a given condition $y$ (e.g., class, text prompt). To this end, the diffusion process is conditioned on $y$. This is typically achieved via classifier-free guidance (CFG) (Ho & Salimans, 2022), where the conditioned prediction $\epsilon_\phi(\mathbf{z}_t; y, t)$ of the noise is extrapolated away from the unconditioned prediction $\epsilon_\phi(\mathbf{z}_t; \varnothing, t)$ by an amount controlled by a scalar $s \in \mathbb{R}$:

$$\epsilon_\phi^s(\mathbf{z}_t; y, t) = \epsilon_\phi(\mathbf{z}_t; y = \varnothing, t) + s \left( \epsilon_\phi(\mathbf{z}_t; y, t) - \epsilon_\phi(\mathbf{z}_t; y = \varnothing, t) \right), \quad (1)$$

where $\varnothing$ indicates a null condition (unconditioned). CFG modifies the score function to steer the process towards regions with a higher ratio of conditional density to the unconditional one. However, it has been observed that CFG trades sample fidelity for diversity (Ho & Salimans, 2022).

**Score Distillation Sampling (SDS)** Over the last two years, text-to-image diffusion methods (Rombach et al., 2022; Saharia et al., 2022; Ramesh et al., 2022; Podell et al., 2023) have achieved unprecedented image generation results by incorporating textual encoder outputs as a condition to the diffusion model. These powerful models are trained on billions of text-image pairs, and such extensive data is currently not available for other domains. The recent introduction of Score Distillation Sampling (SDS) (Poole et al., 2022; Wang et al., 2023a) enables leveraging the priors of pre-trained text-to-image models to facilitate text-conditioned generation in other domains, particularly 3D content generation.

Specifically, given a pretrained diffusion model $\epsilon_\phi$, SDS optimizes a set of parameters $\theta$ of a differentiable parametric image generator $g$, using the gradient of the loss $\mathcal{L}_{\text{SDS}}$ with respect to $\theta$:

$$\nabla_\theta \mathcal{L}_{\text{SDS}} = w(t) \left( \epsilon_\phi^s(\mathbf{z}_t(\mathbf{x}); y, t) - \epsilon \right) \frac{\partial \mathbf{x}}{\partial \theta}, \tag{2}$$

where $\mathbf{x} = g(\theta)$ is an image rendered by $\theta$, $\mathbf{z}_t(\mathbf{x})$ is obtained by adding a Gaussian noise $\epsilon$ to $\mathbf{x}$ corresponding to the $t$-th timestep of the diffusion process, and $y$ is a condition to the diffusion model. In practice, at every optimization iteration, different values of $t$ and Gaussian noise $\epsilon$ are randomly drawn. The parameters $\theta$ are then optimized by computing the gradient of $\mathcal{L}_{\text{SDS}}$ with respect to $\mathbf{x}$ and backpropagating this gradient through the differentiable parametric function $g$.

Poole et al. (2022) formally show that $\mathcal{L}_{\text{SDS}}$ minimizes the KL divergence between a family of Gaussian distributions around $\mathbf{x}$ and the distributions $p(\mathbf{z}_t, y, t)$ learned by the pretrained diffusion model. Intuitively, Equation 2 can be interpreted as follows: since $\mathbf{x} = g(\theta)$ is a clean rendered image, Gaussian noise is first added to it in order to approximately project it to the manifold of noisy images corresponding to timestep $t$. Next, the score $\epsilon_\phi^s(\mathbf{z}_t(\mathbf{x}); y, t)$ provides the direction in which this noised version of $\mathbf{x}$ should be moved towards a denser region in the distribution of real images (noised to timestep $t$ and aligned with the condition $y$). Finally, before the resulting direction can be used to optimize $\theta$, the initially added noise $\epsilon$ is subtracted. We interpret this last step as an attempt to adapt the direction back to the domain of clean rendered images.

While SDS provides an elegant mechanism for leveraging pretrained text-to-image models, SDS-generated results often suffer from oversaturation and lack of fine realistic details. These issues were, in part, attributed to the use of a high CFG value (Wang et al., 2023b), which Poole et al. (2022) empirically found to be necessary to obtain their results. Several derivative approaches have emerged to address these challenges (Metzer et al., 2022; Lin et al., 2023; Chen et al., 2023; Wang et al., 2023b; Huang et al., 2023).

One effective approach for improving the generation quality is time annealing, which gradually reduces the diffusion timesteps $t$ drawn by the optimization process, as it progresses (Lin et al., 2023; Zhu & Zhuang, 2023; Wang et al., 2023b; Huang et al., 2023). Recently, VSD (Wang et al., 2023b) and HiFA (Zhu & Zhuang, 2023) reformulated the distillation loss. HiFA uses a denoised image version instead of the noise prediction, while VSD offers a variational approach, matching the prediction of noisy real images with that of the noisy rendered images via an additional fine-tuned diffusion model. In image editing, DDS (Hertz et al., 2023) observed artifacts when applying SDS to edit real images, which was attributed to a bias in SDS. To mitigate this bias, DDS employs a subtraction of two SDS terms.

In the next section we propose our novel interpretation of SDS via decomposition of the predicted score function into three interpretable components. The insights gained from this decomposition lead us to propose a simple yet effective improvement to SDS, which we call Noise-Free Score Distillation (NFSD), in Section 4. Furthermore, this decomposition enables a simple and unified interpretation of the recent progress in SDS, as discussed in Section 5.

## 3 SCORE DECOMPOSITION

As discussed earlier, the noise predicted by a trained diffusion model aims to be proportional to the score function $\nabla_{\mathbf{z}_t} \log p_t(\mathbf{z}_t)$, where $p_t$ is the marginal distribution of the samples noised to time $t$.

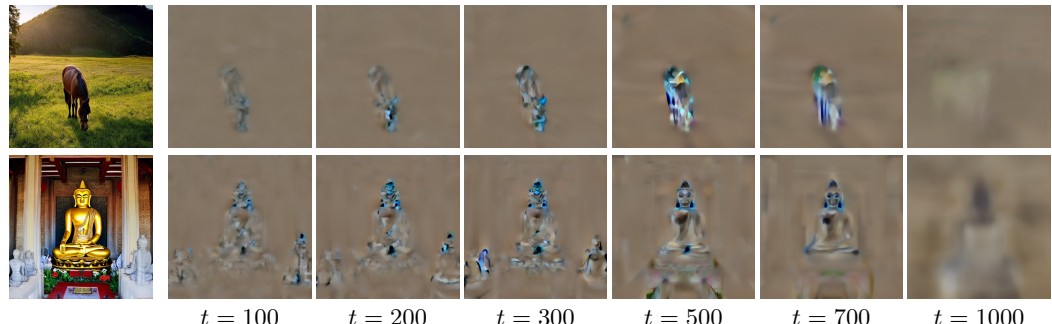

$t = 100$ $\quad$ $t = 200$ $\quad$ $t = 300$ $\quad$ $t = 500$ $\quad$ $t = 700$ $\quad$ $t = 1000$

Figure 2: Visualization of $\delta_{\mathrm{C}}$. The images in the left column are generated by Stable Diffusion (SD version 2.1-base) with the prompts "A photo of a horse in a meadow" and "A statue of Buddha". The other columns visualize $\delta_{\mathrm{C}}$ for added noise $\epsilon$ with magnitude corresponding to different diffusion timesteps $t$. As can be seen, $\delta_{\mathrm{C}}$ is fairly clean and concentrated around the main object in the image. (The visualization is done by decoding each $\delta_{\mathrm{C}}$ using the VAE decoder of SD, please refer to Appendix A.1 for more details).

In order to gain a better understanding of SDS, it is helpful to examine the decomposition of the score direction into several intuitively interpretable components.

First, consider the difference $\delta_{\mathrm{C}} = \epsilon_\phi(\mathbf{z}_t; y, t) - \epsilon_\phi(\mathbf{z}_t; \varnothing, t)$ in Equation 1. While $\epsilon_\phi(\mathbf{z}_t; y, t)$ ideally points towards a local maximum in the probability density of noisy real images conditioned on $y$, $\epsilon_\phi(\mathbf{z}_t; \varnothing, t)$ points towards a denser region in the distribution of unconditioned noisy images. Thus, the difference $\delta_{\mathrm{C}}$ between the two predictions may be thought of as the direction that steers the generated image towards alignment with the condition $y$, and we henceforth refer to it as the *condition direction*.

The condition direction $\delta_{\mathrm{C}}$ is empirically observed to be uncorrelated with the added noise $\epsilon$ and having its significant magnitudes around the condition-specific image regions. As demonstrated in Figure 2, $\delta_{\mathrm{C}}$ is consistently aligned with the condition $y$ for noise corresponding to different timesteps $t$ of the diffusion process. This observation is consistent with the inspiration behind CFG (Ho & Salimans, 2022): classifier-guidance of an implicit classifier $\nabla_{\mathbf{z}_t} \log p^i(y|\mathbf{z}_t)$. Extending this rationale, such a classifier, trained on noisy data $\mathbf{z}_t$, should be invariant to the additive noise $\epsilon$, and its gradients with respect to the input image $\mathbf{z}_t$ should focus on details in $\mathbf{z}_t$ that are most relevant to $y$.

Rewriting Equation 1 using the condition direction $\delta_{\mathrm{C}}$ defined above, we obtain:

$$\epsilon_\phi^s(\mathbf{z}_t; y, t) = \epsilon_\phi(\mathbf{z}_t; \varnothing, t) + s(\epsilon_\phi(\mathbf{z}_t; y, t) - \epsilon_\phi(\mathbf{z}_t; \varnothing, t)) = \epsilon_\phi(\mathbf{z}_t; \varnothing, t) + s\delta_{\mathrm{C}}. \qquad (3)$$

By the nature of its training, the unconditional term $\epsilon_\phi(\mathbf{z}_t; \varnothing, t)$ is expected to predict the noise $\epsilon$ that was added to an image $\mathbf{x} \sim p_{\mathrm{data}}$ to produce $\mathbf{z}_t$. However, in SDS, $\mathbf{z}_t$ is obtained by adding noise to an out-of-distribution (OOD) rendered image $\mathbf{x} = g(\theta)$, which is not sampled from $p_{\mathrm{data}}$. Thus, we can think of $\epsilon_\phi(\mathbf{z}_t; \varnothing, t)$ as a combination of two components, $\epsilon_\phi(\mathbf{z}_t; \varnothing, t) = \delta_{\mathrm{D}} + \delta_{\mathrm{N}}$, where $\delta_{\mathrm{D}}$ is the *domain correction* induced by the difference between the distributions of rendered and real images, while $\delta_{\mathrm{N}}$ is the *denoising direction*, pointing towards a cleaner image. Intuitively, we expect $\delta_{\mathrm{D}}$ to be correlated with the content of $\mathbf{x}(\theta)$, while no such correlation is expected for $\delta_{\mathrm{N}}$.

We are not aware of any general way to explicitly separate $\epsilon_\phi(\mathbf{z}_t; \varnothing, t)$ into these two components. Nevertheless, we attempt to isolate the two components for visualization purposes in Figure 3. The idea is to examine the difference between two unconditional predictions $\epsilon_\phi(\mathbf{z}_t(\mathbf{x}_{\mathrm{ID}}); \varnothing, t)$ and $\epsilon_\phi(\mathbf{z}_t(\mathbf{x}_{\mathrm{OOD}}); \varnothing, t)$, where $\mathbf{z}_t(\mathbf{x}_{\mathrm{ID}})$ and $\mathbf{z}_t(\mathbf{x}_{\mathrm{OOD}})$ are noised in-domain and out-of-domain images, respectively, that depict the same content and are added the same noise $\epsilon$. Intuitively, while $\epsilon_\phi(\mathbf{z}_t(\mathbf{x}_{\mathrm{OOD}}); \varnothing, t)$ both removes noise ($\delta_{\mathrm{N}}$) and steers the sample towards the model's domain ($\delta_{\mathrm{D}}$), the prediction $\epsilon_\phi(\mathbf{z}_t(\mathbf{x}_{\mathrm{ID}})$ mostly just removes noise ($\delta_{\mathrm{N}}$), since the image is already in-domain. Thus, in Figure 3 we use the latter to visualize $\delta_{\mathrm{N}}$ (column (c)) and the difference between the two predictions to visualize $\delta_{\mathrm{D}}$ (column (d)), effectively assuming that $\delta_{\mathrm{N}}$ is shared between $\mathbf{x}_{\mathrm{ID}}, \mathbf{x}_{\mathrm{OOD}}$. As can be seen, $\delta_{\mathrm{N}}$ indeed appears to consist of noise uncorrelated with the image content, while $\delta_{\mathrm{D}}$ is large in areas where the distortion is most pronounced and adding $\delta_{\mathrm{D}}$ to $\mathbf{x}_{\mathrm{OOD}}$ effectively enhances the realism of the image (column (e)). More details about this process can be found in the appendix.

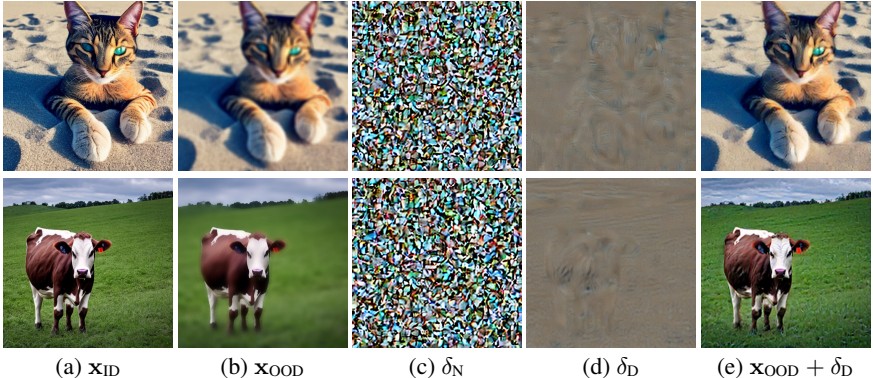

| (a) $\mathbf{x}_{\text{ID}}$ | (b) $\mathbf{x}_{\text{OOD}}$ | (c) $\delta_{\text{N}}$ | (d) $\delta_{\text{D}}$ | (e) $\mathbf{x}_{\text{OOD}} + \delta_{\text{D}}$ |

Figure 3: Visualization of $\delta_{\text{N}}$ and $\delta_{\text{D}}$ (at $t = 400$). Columns (a) and (b) show a pair of in-domain ($\mathbf{x}_{\text{ID}}$) and out-of-domain ($\mathbf{x}_{\text{OOD}}$) images, both depicting the same underlying content. We add the same noise to both images, and use the pre-trained diffusion model to predict the score. Intuitively, the noised $\mathbf{x}_{\text{ID}}$ image requires no domain correction, and thus the predicted score consists of only $\delta_{\text{N}}$, shown in (c). Subtracting $\delta_{\text{N}}$ from the prediction for the noised $\mathbf{x}_{\text{OOD}}$ image gives us the domain correction $\delta_{\text{D}}$, shown in (d). Indeed, adding $\delta_{\text{D}}$ to $\mathbf{x}_{\text{OOD}}$ produces a more realistic image (e).

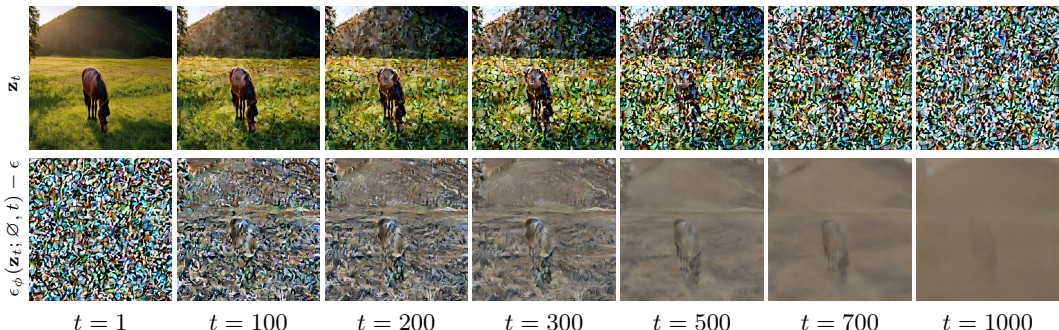

| $t = 1$ | $t = 100$ | $t = 200$ | $t = 300$ | $t = 500$ | $t = 700$ | $t = 1000$ |

Figure 4: Visualization of $\delta_{\text{N}} - \epsilon$. Top row: noise $\epsilon$ corresponding to different diffusion timesteps $t$ is added to an in-domain image of a horse (as indicated below each column). Bottom row: the residual $\epsilon_\phi(\mathbf{z}_t; \varnothing, t) - \epsilon$ between the network prediction and the actual noise. Since the original image is in-domain (generated by SD), $\delta_{\text{D}} \approx 0$, and therefore, $\epsilon_\phi(\mathbf{z}_t; \varnothing, t) \approx \delta_{\text{N}}$. For visualization purposes, the residual is decoded and clamped between -1 and 1. Although we do not expect the residual $\delta_{\text{N}} - \epsilon$ to be correlated with the image, it may be seen that some correlation in fact exists, and furthermore, the residual becomes progressively noisier at smaller timesteps $t$.

To summarize so far, using the components discussed above, we can rewrite the CFG score as:

$$\epsilon_\phi^s(\mathbf{z}_t; y, t) = \delta_{\text{D}} + \delta_{\text{N}} + s\delta_{\text{C}}. \tag{4}$$

Poole et al. (2022) define the SDS loss using the difference between the CFG score and the noise $\epsilon$ that was added to the rendered image $\mathbf{x}$ to produce $\mathbf{z}_t$, i.e.,

$$\nabla_\theta \mathcal{L}_{\text{SDS}} = w(t)(\epsilon_\phi^s(\mathbf{z}_t; y, t) - \epsilon)\frac{\partial \mathbf{x}}{\partial \theta} = w(t)(\delta_{\text{D}} + \delta_{\text{N}} + s\delta_{\text{C}} - \epsilon)\frac{\partial \mathbf{x}}{\partial \theta}. \tag{5}$$

Note that while both $\delta_{\text{D}}$ and $\delta_{\text{C}}$ are needed to steer the rendered image towards an in-domain image aligned with the condition $y$, the residual $\delta_{\text{N}} - \epsilon$ is generally non-zero and noisy, and this issue becomes increasingly pronounced when smaller time steps, responsible for the formation of fine details, are employed, as visualized in Figure 4. This residual may explain, in part, the lower quality images generated using SDS, compared to ancestral sampling: at each optimization step, the optimized parameters $\theta$ are guided at a random direction depending on $\delta_{\text{N}} - \epsilon$, resulting in an averaging effect. While higher level semantics are roughly less affected, fine and medium level details, from lower diffusion times $t$ tend to be over-smoothed. Previous works (Hertz et al., 2023; Wang et al., 2023b) have also observed that the subtraction of $\epsilon$ indeed leads to blurry results.

Importantly, our decomposition in Equation 5 explains the need for using a large CFG coefficient in SDS (e.g., $s = 100$), as this enables the image-correlated $s\delta_{\text{C}}$ term to dominate the loss, making

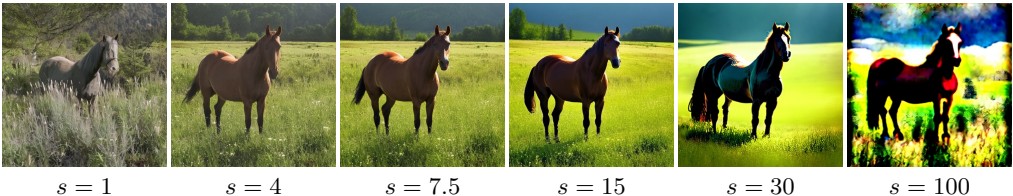

$$s = 1 \qquad s = 4 \qquad s = 7.5 \qquad s = 15 \qquad s = 30 \qquad s = 100$$

Figure 5: The impact of CFG on ancestral sampling. We generate all images using the same prompt "A photo of a horse in a meadow" with the same seed and different values of the CFG parameter $s$. As can be seen, large values of $s$ lead to over-saturated and less realistic results.

the noisy residual $\delta_N - \epsilon$ relatively negligible. However, high CFG coefficients are known to yield less realistic results as demonstrated in Figure 5, typically leading to over-saturated images and NeRFs. Although high CFG coefficients have also been held responsible for lack of result diversity, we demonstrate in Figure 11 that this assertion is not accurate.

## 4  NOISE FREE SCORE DISTILLATION

As discussed above, ideally only the $s\delta_C$ and the $\delta_D$ terms should be used to guide the optimization of the parameters $\theta$. While $\delta_C$ is simply the difference between the conditioned and the null-conditioned predictions, $\delta_D$ is more challenging to separate from $\delta_N$, as they are both part of the predicted noise $\epsilon_\phi(\mathbf{z}_t; \varnothing, t)$.

To extract $\delta_D$, we distinguish between different stages in the backward (denoising) diffusion process. First, note that the noise variance, i.e., the magnitude of the noise to be removed, is monotonically decreased in the backward process. Thus, for sufficiently small timestep values $t$, $\delta_N$ becomes small enough to be neglected, and the score $\epsilon_\phi(\mathbf{z}_t; \varnothing, t) = \delta_N + \delta_D$ is approximately $\delta_D$. Specifically, we apply this approximation for $t < 200$.

As for the larger timestep values, $t \geq 200$, we propose to approximate $\delta_D$ by the difference $\epsilon_\phi(\mathbf{z}_t; \varnothing, t) - \epsilon_\phi(\mathbf{z}_t; y = p_{\text{neg}}, t)$, where $p_{\text{neg}}$ = "unrealistic, blurry, low quality, out of focus, ugly, low contrast, dull, dark, low-resolution, gloomy". Here, we are making the assumption that $\delta_{C=p_{\text{neg}}} \approx -\delta_D$, and thus $\epsilon_\phi(\mathbf{z}_t; \varnothing, t) - \epsilon_\phi(\mathbf{z}_t; y = p_{\text{neg}}, t) = -\delta_{C=p_{\text{neg}}} \approx \delta_D$.

To conclude, we approximate $\delta_D$ by

$$\delta_D = \begin{cases} \epsilon_\phi(\mathbf{z}_t; \varnothing, t), & \text{if } t < 200 \\ \epsilon_\phi(\mathbf{z}_t; \varnothing, t) - \epsilon_\phi(\mathbf{z}_t; y = p_{\text{neg}}, t), & \text{otherwise,} \end{cases} \qquad (6)$$

and use the resulting $\delta_C$ and $\delta_D$ to define an alternative, *noise-free score distillation* loss $\mathcal{L}_{\text{NFSD}}$, whose gradients are used to optimize the parameters $\theta$, instead of $\nabla_\theta \mathcal{L}_{\text{SDS}}$:

$$\nabla_\theta \mathcal{L}_{\text{NFSD}} = w(t) \left( \delta_D + s\delta_C \right) \frac{\partial \mathbf{x}}{\partial \theta}, \qquad (7)$$

In Figure 7 and Figure 8 we show that these seemingly small changes in the definition of the loss lead to a noticeable improvement in the quality of generated images, as well as NeRFs. Note, that while in the results of SDS we use $s = 100$, in our results we use the commonly used value of $s = 7.5$. The reason for this is that by taking the measures described above to approximately eliminate the $\delta_N$ component, it is no longer necessary to resort to a large value of $s$ to make the $\delta_C$ term dominant.

## 5  DISCUSSION

Our score decomposition formulation can be used to explain previous works that were proposed to improve the SDS loss. This demonstrates the wide scope and applicability of our formulation.

**DDS.** Hertz et al. (2023) propose an adaptation of the SDS loss for image editing task. Specifically, instead of randomly initializing the optimization process as in SDS, it is initialized with the (in-domain) input image. DDS optimizes the input image according to the text condition, while preserving image attributes that are irrelevant to the edit task guided by the input prompt. The gradients used by DDS are defined by

$$\nabla_\theta \mathcal{L}_{\text{DDS}} = \nabla_\theta \mathcal{L}_{\text{SDS}}(\mathbf{z}_t(\mathbf{x}), y) - \nabla_\theta \mathcal{L}_{\text{SDS}}(\tilde{\mathbf{z}}_t(\tilde{\mathbf{x}}), \tilde{y}), \qquad (8)$$

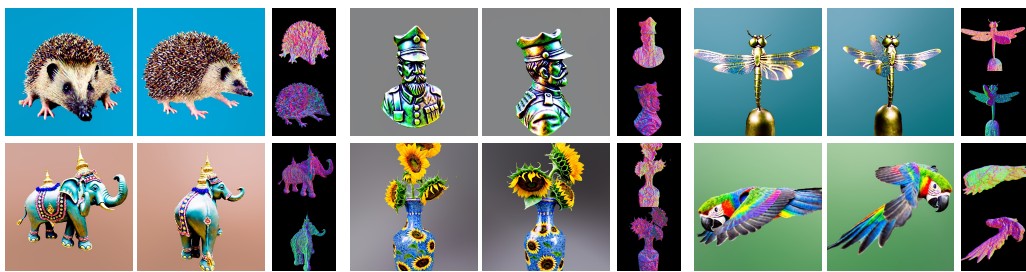

Figure 6: NeRFs optimized with NFSD.

where $\tilde{\mathbf{z}}_t(\tilde{\mathbf{x}}), \tilde{y}$ denote the noisy original input image and its corresponding prompt, respectively. Here $y$ denotes the prompt that describes the edit, and $\mathbf{x}, \tilde{\mathbf{x}}$ are noised with the same noise $\epsilon$. Incorporating our score decomposition into Equation 8 yields

$$\nabla_\theta \mathcal{L}_{\text{DDS}} = w(t)(\delta_D + \delta_N + s\delta_{C_{\text{edit}}} - \epsilon) - w(t)(\delta_D + \delta_N + s\delta_{C_{\text{orig}}} - \epsilon) = w(t)s(\delta_{C_{\text{edit}}} - \delta_{C_{\text{orig}}}). \quad (9)$$

Our formulation helps to understand the high-quality results achieved by DDS: the residual component which makes the results in SDS over-smoothed and over-saturated is cancelled out. Moreover, since the optimization is initialized with an in-domain image, the $\delta_D$ component is not effectively needed and cancelled out. The remaining direction is the one relevant to the difference between the original prompt and the new one.

**ProlificDreamer.** Wang et al. (2023b) tackle the generation task, and propose the VSD loss, which successfully alleviates the over-smoothed and over-saturated results obtained by SDS. In VSD, alongside the pretrained diffusion model $\epsilon_\phi$, another diffusion model $\epsilon_{\text{LoRA}}$ is trained during the optimization process. The $\epsilon_{\text{LoRA}}$ model is initialized with the weights of $\epsilon_\phi$, and during the optimization process it is fine-tuned with rendered images $\mathbf{x} = g(\theta)$. Effectively, the rendered images during the optimization are out-of-domain for the original pretrained model distribution, but are in-domain for $\epsilon_{\text{LoRA}}$. Hence, the gradients of the VSD loss are defined as

$$\nabla_\theta \mathcal{L}_{\text{VSD}} = w(t)\left(\epsilon_\phi^s(\mathbf{z}_t(\mathbf{x}); y, t) - \epsilon_{\text{LoRA}}(\mathbf{z}_t(\mathbf{x}); y, t, c\right)\frac{\partial \mathbf{x}}{\partial \theta}, \quad (10)$$

where $c$ is another condition that is added to $\epsilon_{\text{LoRA}}$ and represents the camera viewpoint of the rendered image $\mathbf{x}$. Viewed in terms of our score decomposition, since $\epsilon_{\text{LoRA}}$ is fine-tuned on $\mathbf{x}$, both $\delta_C$ and $\delta_D$ are approximately 0, thus it simply predicts $\delta_N$. Therefore, $\nabla_\theta \mathcal{L}_{\text{VSD}}$ can be written as

$$\nabla_\theta \mathcal{L}_{\text{VSD}} = w(t)(\delta_D + \delta_N + s\delta_C - \delta_N)\frac{\partial \mathbf{x}}{\partial \theta} = w(t)(\delta_D + s\delta_C)\frac{\partial \mathbf{x}}{\partial \theta}, \quad (11)$$

i.e., it approximates exactly the same terms as our NFSD. It should be noted that unlike our approach, VSD has a considerable computational overhead of fine-tuning the additional diffusion model during the optimization process.

## 6 EXPERIMENTS

We implement NFSD using the threestudio (Guo et al., 2023) framework for text-based 3D generation. Unless stated otherwise, all 3D models are optimized for $25,000$ iterations using AdamW optimizer (Loshchilov & Hutter, 2017) with a learning rate of $0.01$. The initial rendering resolution of $64 \times 64$ is increased to $512 \times 512$ after $5,000$ iterations; at the same time we anneal the maximum diffusion time to $500$ as proposed by Lin et al. (2023); Wang et al. (2023b). The implicit volume is initialized according to the object-centric initialization (Lin et al., 2023; Wang et al., 2023b). We alternate the background between random solid-color and a learned neural environment map. The pre-trained text-to-image diffusion model for all experiments is Stable Diffusion 2.1-base (Rombach et al., 2022), a latent diffusion model with $\epsilon$-prediction.

**3D generation.** Figure 6 showcases several NeRFs optimized using our NFSD. As can be seen the rendered images are sharp and contain highly intricate details. The prompts used and additional examples can be found in Appendices A.2 and A.7, respectively.

**Comparison with SDS.** We compare our NFSD with SDS under different parametric generators and different configurations. In each comparison, the same seed is used by both methods. Specifically,

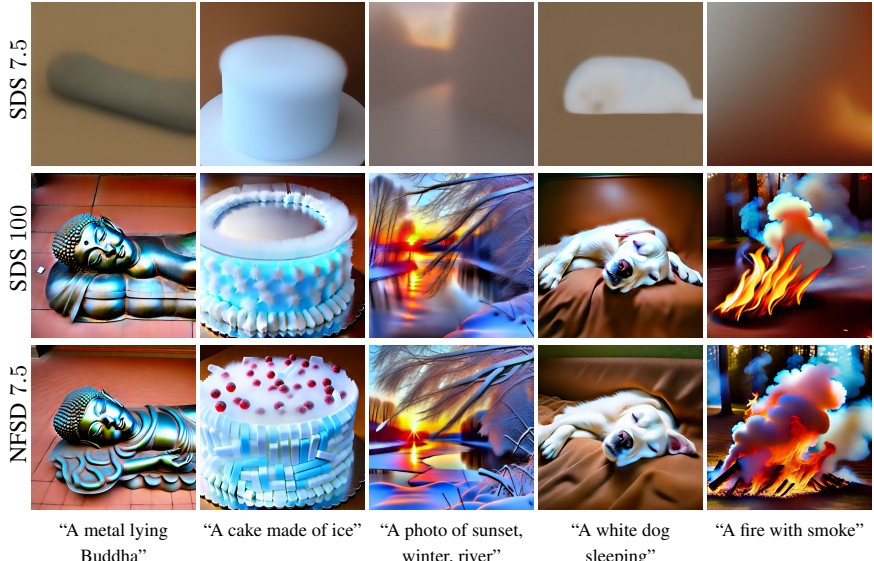

Figure 7: 2D image generation with SDS and NFSD. We directly optimize the latent space of SD-2.1-base (Rombach et al., 2022). Top row: SDS with CFG of 7.5 generates overly-smooth images that severely lack detail. The common solution is to increase the CFG scale to 100 (middle row). The high CFG enables a reasonable distillation of the score, but some artifacts remain (e.g., the dog's purple eyes, the frosting in the cake) and realistic details are still lacking (e.g., the fire). Bottom row: using the standard CFG of 7.5, our NFSD succeeds in distilling finer details, such as the frosting on the cake or the fire example.

we set the same seed for the guidance process, i.e., the noise and the diffusion time are the same for both methods, at every step of the optimization process.

*2D image generation:* Here, we directly optimize the latent code of Stable Diffusion, a $64 \times 64 \times 4$ tensor. In the notations defined in Section 2, $\theta \in \mathbb{R}^{64 \times 64 \times 4}$ and $g(\theta) = \theta$. To initialize the optimization process, $\theta$ is sampled from a Gaussian Distribution. We then use either $\mathcal{L}_{\text{NFSD}}$ or $\mathcal{L}_{\text{SDS}}$ as the only loss for $1,000$ iterations. As illustrated in Figure 7, SDS optimization with a nominal CFG scale (7.5) yields over-smoothed images, while using a high CFG scale (100) generates the main object but lacks background details and occasionally introduces aritfacts. In contrast, NFSD optimization is able to produce more pleasing results in which the object is clear, the background is detailed, and the image looks more realistic, even when using a CFG value of 7.5. For example, observe the fur of the dog which contains artifacts in the SDS-100 configuration, and looks more realistic with NFSD-7.5. In addition, observe the fire flames which feature fewer details and therefore look painted with SDS-100, and exhibit more detail and realism when using NFSD-7.5.

*Text-to-NeRF synthesis:* In Figure 8 we use SDS and NFSD to optimize NeRFs given text prompts. Here we do not preform time annealing of the diffusion, which hinders a bit the quality of the output model, but allows better emphasis on the differences between SDS and NFSD. As can be seen, although SDS succeeds in generating plausible 3D objects, it typically generates fewer fine details. Note for example the wings of the eagle and the mane of the horse. Note that, unlike images, the NeRF representation is inherently smooth; furthermore, additional losses, such as shape density, are applied. These regularizations, combined with a high CFG value, enable SDS to produce plausible results, despite the unwanted noise distillation. In contrast, NFSD produces more detailed results with ordinary CFG values by attempting to eliminate the noise and explicitly approximate $\delta_{\text{D}}$.

**Comparison with related methods.** In Figure 9 we compare our method with recent approaches including DreamFusion (Poole et al., 2022), Magic3D (Lin et al., 2023), Latent-NeRF (Metzer et al., 2022), Fantasia3D (Chen et al., 2023), and ProlificDreamer (Wang et al., 2023b). Following previous works (Metzer et al., 2022; Wang et al., 2023b; Chen et al., 2023; Lin et al., 2023), for each of these methods we compare to results that were reported by the authors. As can be seen, our method achieves comparable or better results, while being significantly simpler than most of these methods. For example, observe the roof of the cottage, which is highly detailed in our method and in ProlificDreamer, but exhibits less detail in the other methods. Unlike ProlificDreamer which requires

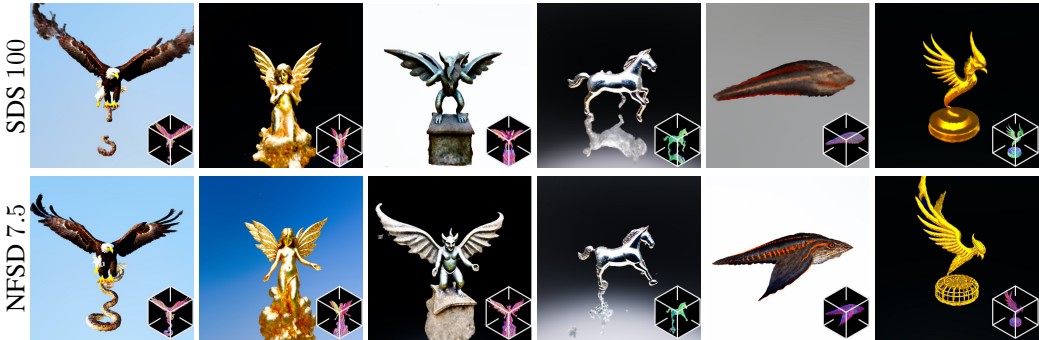

Figure 8: Comparison of NeRFs optimized by SDS (top) and NFSD (bottom). NFSD allows better distillation: e.g., observe the wings of the eagle (leftmost), the angel's stomach (second from left), and the horse's mane and tail (third from right). Furthermore, NFSD creates fewer sporadic features, as in the gargoyle (third from left). The prompts are reported in Appendix A.2.

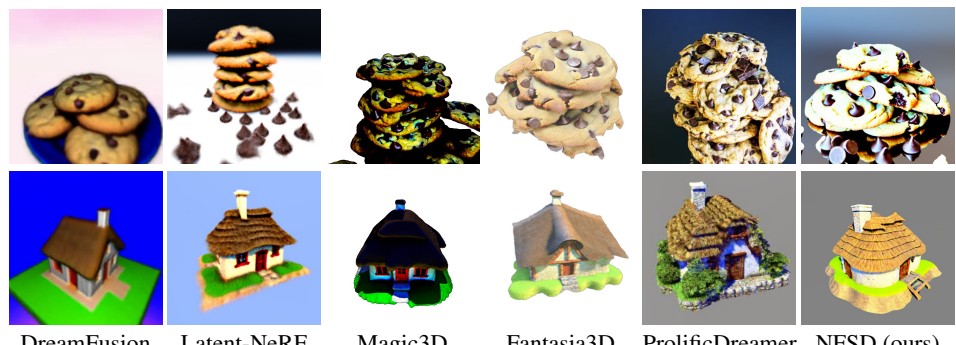

| DreamFusion | Latent-NeRF | Magic3D | Fantasia3D | ProlificDreamer | NFSD (ours) |

Figure 9: Comparison of NFSD with different methods. Our NFSD is of high resolution and exhibits detailed features. In the top row we used the prompt "A plate piled high with chocolate chip cookies", and in the bottom one we used "A 3D model of an adorable cottage with a thatched roof".

optimizing a diffusion model in parallel to the NeRF optimization, our method is simpler to implement and the optimization process is much faster. Please note that the original methods differ in their implementation details, including the diffusion model that guides the optimization. Therefore, in Appendix A.8, we provide comparison using threestudio (Guo et al., 2023) for all method results.

## 7  CONCLUSION AND FUTURE WORK

In this paper, we have revisited the SDS process and introduced a novel interpretation, dissecting the score into three distinct components: the condition, the domain, and the denoising components. Through this novel perspective, we proposed a simple distillation process, which we refer to as Noise-Free Score Distillation (NFSD). NFSD was developed with the explicit goal of preventing noise distillation during the optimization process. Notably, NFSD requires only minimal adjustments to the SDS framework, all while operating with a nominal Classifier-Free Guidance (CFG) scale. Despite its simplicity, NFSD has shown promising potential in advancing the generation of 3D objects, demonstrating notable improvements when compared to both SDS and existing approaches.

While NFSD enables better score distillation compared with SDS, two main drawbacks of the SDS process still exist, namely the well known Janus problem (multi-face) and low diversity. We believe that the latter is a direct result of the distillation process mechanism: the diffusion scores that guide the optimization are uncorrelated across successive iterations, even if noise is successfully eliminated. While annealing the diffusion time as the optimization progresses is helpful, designing a more principled noise scheduling might prove more effective, and lead to improved diversity.

Additionally, we recognize the challenging nature of evaluating NeRFs generated from text, with a current absence of suitable metrics and benchmarks to assess NeRF quality and enable comprehensive comparisons across various methodologies. We believe that the development of such metrics is essential for advancing research in this domain and and would like to investigate this in the future.

ACKNOWLEDGMENTS

We thank Rinon Gal and Yuval Alaluf for their feedback and helpful suggestions. This work was supported in part by the Israel Science Foundation (grants no. 2492/20, 3611/21 and 3441/21), Len Blavatnik and the Blavatnik family foundation.

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

# A APPENDIX

## A.1 VISUALIZATION OF SCORE COMPONENTS

In Figures 2, 3 and 4 we visualize the score components, or manipulation of them. Stable Diffusion operates in the latent space of a pretrained autoencoder, and therefore we use the decoder to obtain these images. At a first glance it may be surprising that the decoder can decode such latents, as they are out of the distribution that it was trained on. However, as shown in (Turner, 2022) and discussed by (Metzer et al., 2022), the RGB value at a certain pixel, can be approximated by applying a fixed linear transformation on the corresponding pixel of the latent code. Hence, we conclude that the latent space of Stable Diffusion behaves similarly to the RGB space, and decoding to RGB latent codes that are out of distribution is still meaningful. We also note that when decoding the zero latent code, we get the brown RGB image shown on the right. Therefore, this color in our visualizations indicates a value of $0$ in the latent code.



Additionally, in Figure 3 we visualize $\delta_D$ and $\delta_N$ by generating pairs of $\mathbf{x}_{ID}$ (in-domain) and $\mathbf{x}_{OOD}$ (out-of-domain) images. Additional components are visualized in Figure 10. The two ID images were generated using DDIM sampling with the prompts "a pretty cat on a sand" and "a cow in a field". Subsequently, generate the corresponding OOD images by applying a Gaussian filter to the cat image and a bilateral filter to the cow image.

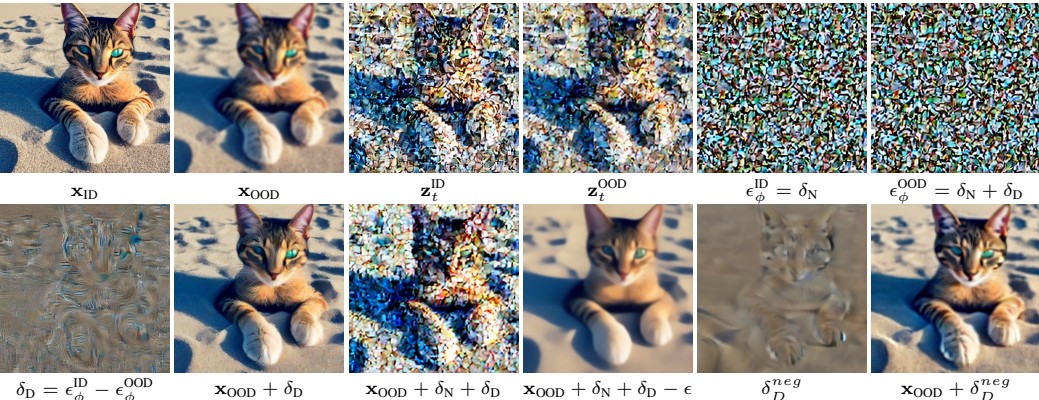

Figure 10: Extension of Figure 3 in the main paper. Top row: we generate a pair of in-domain ($\mathbf{x}_{ID}$) and out-of-domain ($\mathbf{x}_{OOD}$) images, both depicting the same underlying content. The same noise $\epsilon$ is added to both images yielding $\mathbf{z}_t^{ID} = \mathbf{z}_t(\mathbf{x}_{ID})$ and $\mathbf{z}_t^{OOD} = \mathbf{z}_t(\mathbf{x}_{OOD})$, for ID and OOD respectively. We then calculate the predicted score of the pre-trained diffusion model, $\epsilon_\phi^{ID} = \epsilon_\phi(\mathbf{z}_t(\mathbf{x}_{ID}); \varnothing, t)$ and $\epsilon_\phi^{OOD} = \epsilon_\phi(\mathbf{z}_t(\mathbf{x}_{OOD}); \varnothing, t)$. Bottom row: we first calculate $\delta_D$ by subtracting the OOD score prediction from the ID score prediction, and then show that by adding this component to the OOD image, the quality of the resulting image, $\mathbf{x}_{OOD} + \delta_D$, is significantly improved. Conversely, when adding both $\delta_D$ and $\delta_N$ to $\mathbf{x}_{OOD}$ (resulting in $\mathbf{x}_{OOD} + \delta_N + \delta_D$), the noise is clearly apparent. When subtracting $\epsilon$ from $\mathbf{x}_{OOD} + \delta_N + \delta_D$, in a manner similar to the procedure in SDS, the resulting image ($\mathbf{x}_{OOD} + \delta_N + \delta_D - \epsilon$) looks blurry. We also show $\delta_D^{neg}$ as defined in Equation 6, and demonstrate that its addition to $\mathbf{x}_{OOD}$ is similar to the result of adding the $\delta_D$ defined in the leftmost column.

## A.2 PROMPTS USED IN THE PAPER

Here we report the prompts used in several of the figures, in a left-to-right, top-to-bottom fashion.

In Figure 6: "A huge Hedgehog", "a soldier iron decor pen holder", "a trunk up statue of an Elephant with Thailand Decoration, side view", "a photo of a vase with sunflowers", "a brass statue of a dragonfly" and "a rainbow colored wings spread parrot".

In Figure 8: "a eagle catching a snake", "a golden statue of a fairy angel with white wings", "a metal gargoyle statue with white wings, "a silver metal running horse on a table glass", "a huge Hystrix" and "a phoenix in golden cage".

## A.3 DIVERSITY OF RESULTS

We explore the diversity of generated results achieved by our method compared to SDS-100. We follow a procedure similar to the one outlined in Sec. 6 by training a 2D-latent image, albeit without implementing any annealing of the diffusion time. As illustrated in Figure 11, both methods exhibit limited variation, portraying similar objects of comparable sizes positioned centrally within the image. Therefore, we infer that the limited diversity observed in SDS cannot be attributed solely to the use of a high CFG scale.

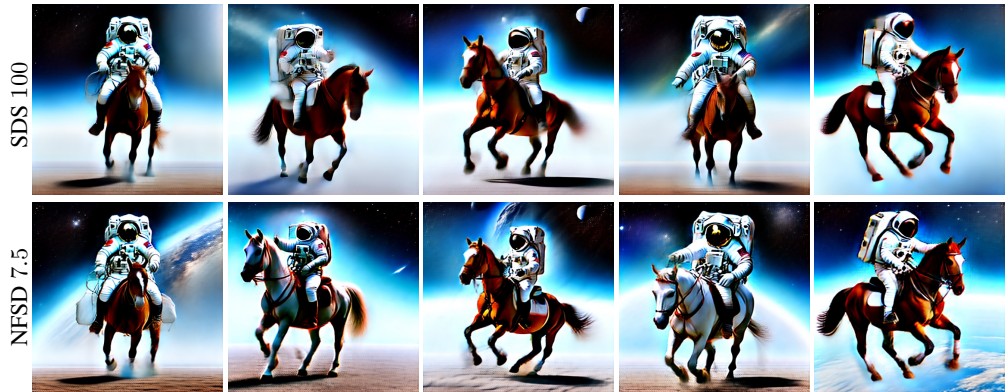

Figure 11: Diversity in NFSD and SDS Results. We optimize 2D-latent images using the prompt "An astronaut riding a horse" for both SDS (top) and NFSD (bottom) with each column originating from a different seed.

## A.4 NEGATIVE PROMPTS

Here we show the relation between the term $\delta_D$ presented in the paper to the technique of using negative prompts. Specifically, the usage of negative prompts, namely replacing the empty unconditional prompt ("") with a selected negative prompt, is well-known to improve the results of ancestral-sampling (Liu et al., 2022). In the context of SDS, it is also possible to use a negative prompt instead of the empty prompt, yielding

$$\nabla_\theta \mathcal{L}_{\text{SDS-neg}} = w(t)(\epsilon_\phi(\mathbf{z}_t; \varnothing, t) + s(\epsilon_\phi(\mathbf{z}_t; y, t) - \epsilon_\phi(\mathbf{z}_t; p_{\text{neg}}, t)) - \epsilon)\frac{\partial \mathbf{x}}{\partial \theta}. \tag{12}$$

Since

$$\epsilon_\phi(\mathbf{z}_t; y, t) - \epsilon_\phi(\mathbf{z}_t; p_{\text{neg}}, t) = (\epsilon_\phi(\mathbf{z}_t; y, t) - \epsilon_\phi(\mathbf{z}_t; \varnothing, t)) - (\epsilon_\phi(\mathbf{z}_t; p_{\text{neg}}, t) - \epsilon_\phi(\mathbf{z}_t; \varnothing, t)), \tag{13}$$

then, $\mathcal{L}_{\text{SDS-neg}}$ can also be written as

$$\nabla_\theta \mathcal{L}_{\text{SDS-neg}} = w(t)(\delta_D + \delta_N + s(\delta_C + \delta_D) - \epsilon)\frac{\partial \mathbf{x}}{\partial \theta}, \tag{14}$$

since $\delta_D$ is defined as $-(\epsilon_\phi(\mathbf{z}_t; p_{\text{neg}}, t) - \epsilon_\phi(\mathbf{z}_t; \varnothing, t))$ in Equation 6. Interestingly, using a negative prompt in SDS, mimics the addition of an amplified $\delta_D$ term to the SDS loss. This further strengthens the importance of the term $\delta_D$, and supports its definition with the negative prompt. As can be seen in Figure 13, the negative prompt (NP) usually adds details to both SDS and NFSD (e.g., the smoke). However, The results obtained by SDS-100+NP sometimes exhibit some artifacts such as the dark spots on the cake's frosting, and above the dog's eyes. NFSD-7.5+NP obtains highly detailed results (e.g., the ice on the berries), but sometimes slightly deviates from the prompt (e.g., the cake does not look made of ice with NP).

## A.5 ABLATION STUDIES

**Time Threshold** We empirically validate our decision to set $t_s = 200$ as the sufficiently small time in Equation 6 for estimating $\epsilon_\phi(\mathbf{z}_t; \varnothing, t) \approx \delta_D$. We follow a similar procedure to the one outlined in Sec. 6 without time annealing. As depicted in Figure 12, a time interval that is too small fails to regulate the generated image, resulting in noticeable artifacts, while too large values of $t_s$ lead to an overly smoothed result. Note, that in all the visualized examples we use a CFG scale of 7.5, and therefore the results for $t_s = 1000$ look worse than those of SDS-100.

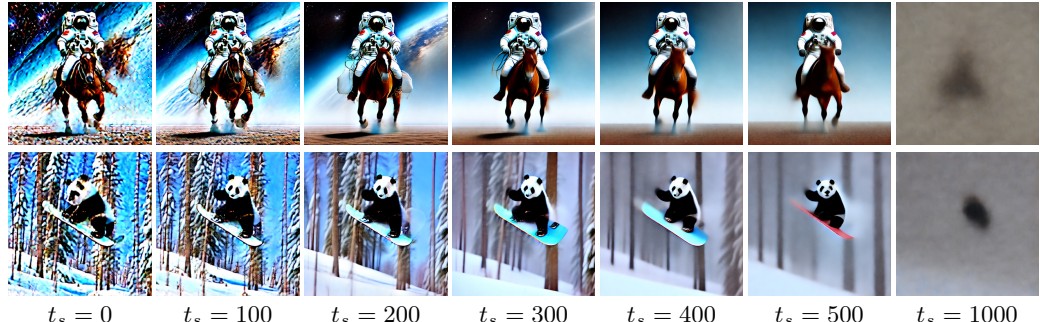

$$t_s = 0 \quad t_s = 100 \quad t_s = 200 \quad t_s = 300 \quad t_s = 400 \quad t_s = 500 \quad t_s = 1000$$

Figure 12: Different time thresholds $t_s$. For every $t < t_s$, $\delta_D$ is estimated via $\epsilon_\phi(\mathbf{z}_t; \varnothing, t)$. The prompts used are: "An astronaut riding a horse" and "Panda snowboarding".

**Ablating $\delta_D$** To better understand the role of $\delta_D$ in NFSD, we remove it from our loss, keeping only the $\delta_C$ term. In the second row of Figure 13, we show results where we optimize a 2D-latent image only with $\delta_C$. As can be seen, optimizing with $\delta_C$ results in an image that follows the prompt, but has many artifacts.

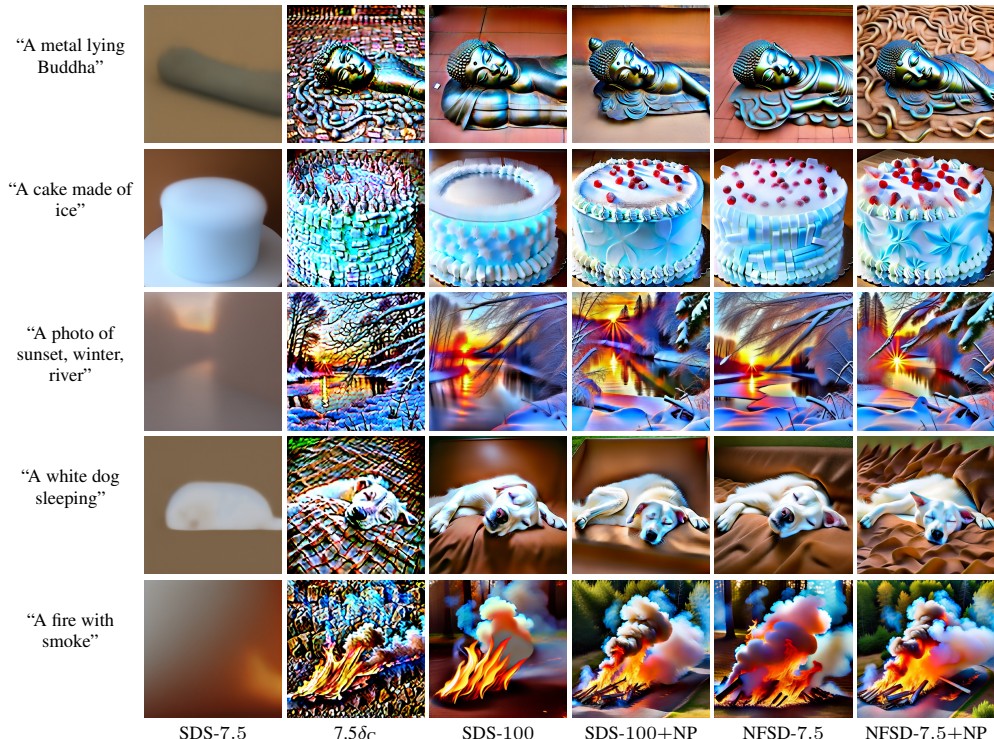

| | SDS-7.5 | $7.5\delta_C$ | SDS-100 | SDS-100+NP | NFSD-7.5 | NFSD-7.5+NP |

Figure 13: Extension of Figure 7. In the second column we present results obtained by using only the $\delta_C$ component in NFSD. As shown, this results in many artifacts. Using the negative prompt adds details to both SDS-100 and NFSD-7.5 as shown in the fourth and rightmost columns for SDS and NFSD, respectively. However, SDS-100+NP exhibits some artifacts (e.g., the cake frosting, the dog's eyes). NFSD-7.5+NP produces sharp and highly detailed results, but sometimes slightly deviates from the prompt (e.g., the cake).

## A.6 QUANTITATIVE EVALUATION

**FID** We quantitatively evaluate both our NFSD and SDS for the task of 2D-image generation using the COCO2014 caption dataset. We have sampled 5K captions and images from the validation dataset. The FID scores between the generated data of each method and the COCO dataset are shown in Table 1. As can be seen in the table, NFSD achieves better FID compared to SDS-100, confirming the better image quality achieved by our method.

|  | SDS-100 | NFSD (ours) |
|---|---|---|
| FID | 39.04 | **35.18** |

Table 1: FID-5K results on COCO2014. NFSD achieves lower FID score compared to SDS.

**User Study** Following Magic3D (Lin et al., 2023) and ProlificDreamer (Wang et al., 2023b), we conducted a user study comparing rendered NeRF images obtained by DreamFusion (Poole et al., 2022), Fantasia3D (Chen et al., 2023), Magic3D (Lin et al., 2023), ProlificDreamer (Wang et al., 2023b), and our NFSD method. Specifically, we used 15 prompts from previous works, and for each prompt, we displayed images corresponding to this prompt obtained by methods that provide a result for this prompt. Respondents were asked to choose the image that is most aligned with the given prompt, and the highest quality image. Deciding between methods may be very difficult in certain examples, and therefore we allowed to choose results from more than a single method. The final score of each method is defined as the percentage of times the results of this method were selected. Note that since we allow respondents to choose more than a single method in each question, the scores are not summed to 100. The results are summarized in Table 2. As can be seen, all methods provide results that are aligned with the prompt and our method attains the highest score. In terms of quality, our method obtains significantly better results compared with the competing methods.

|  | DreamFusion | Fantasia3D | Magic3D | ProlificDreamer | NFSD (ours) |
|---|---|---|---|---|---|
| prompt alignment | 46.09 | 61.74 | 42.03 | 41.55 | **65.51** |
| image quality | 3.19 | 27.83 | 17.75 | 37.68 | **70.14** |

Table 2: User study results. The highest possible score for each method is 100.

### A.7 ADDITIONAL RESULTS

In Figures 14 and 15 we present additional results for text-to-NeRF generation using our NFSD.

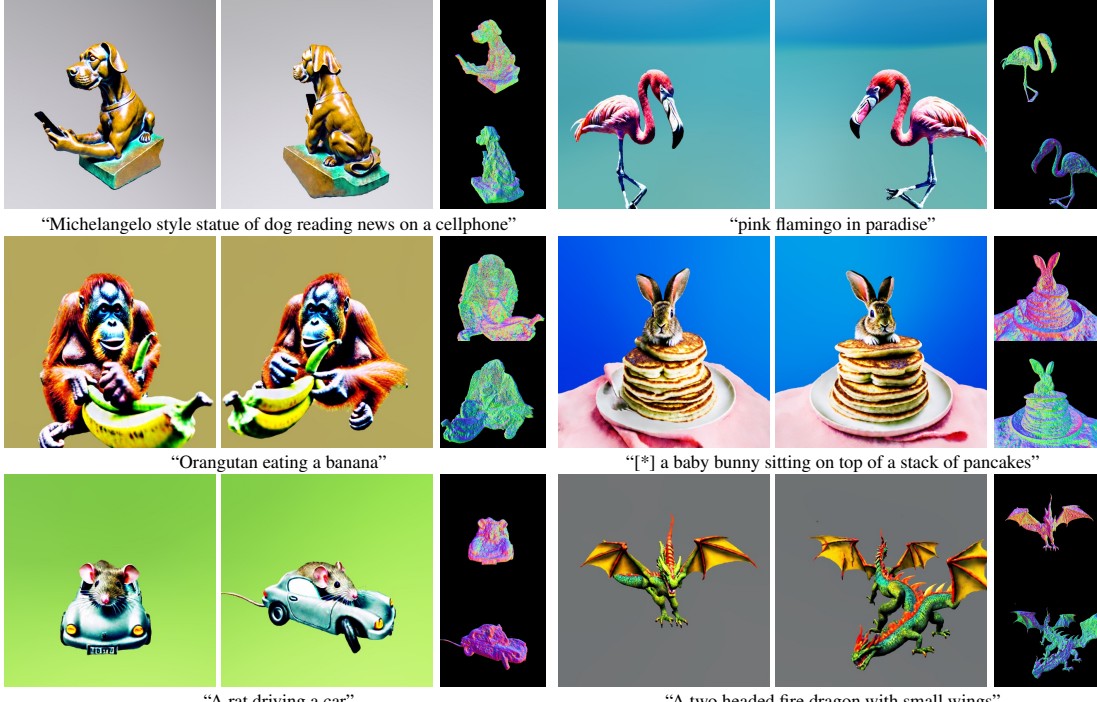

"Michelangelo style statue of dog reading news on a cellphone"  "pink flamingo in paradise"

"Orangutan eating a banana"  "[*] a baby bunny sitting on top of a stack of pancakes"

"A rat driving a car"  "A two headed fire dragon with small wings"

Figure 14: NeRFs optimized with NFSD. [*] "A zoomed out DSLR photo of", [...]-"A wide angle zoomed out DSLR photo of zoomed out view of".

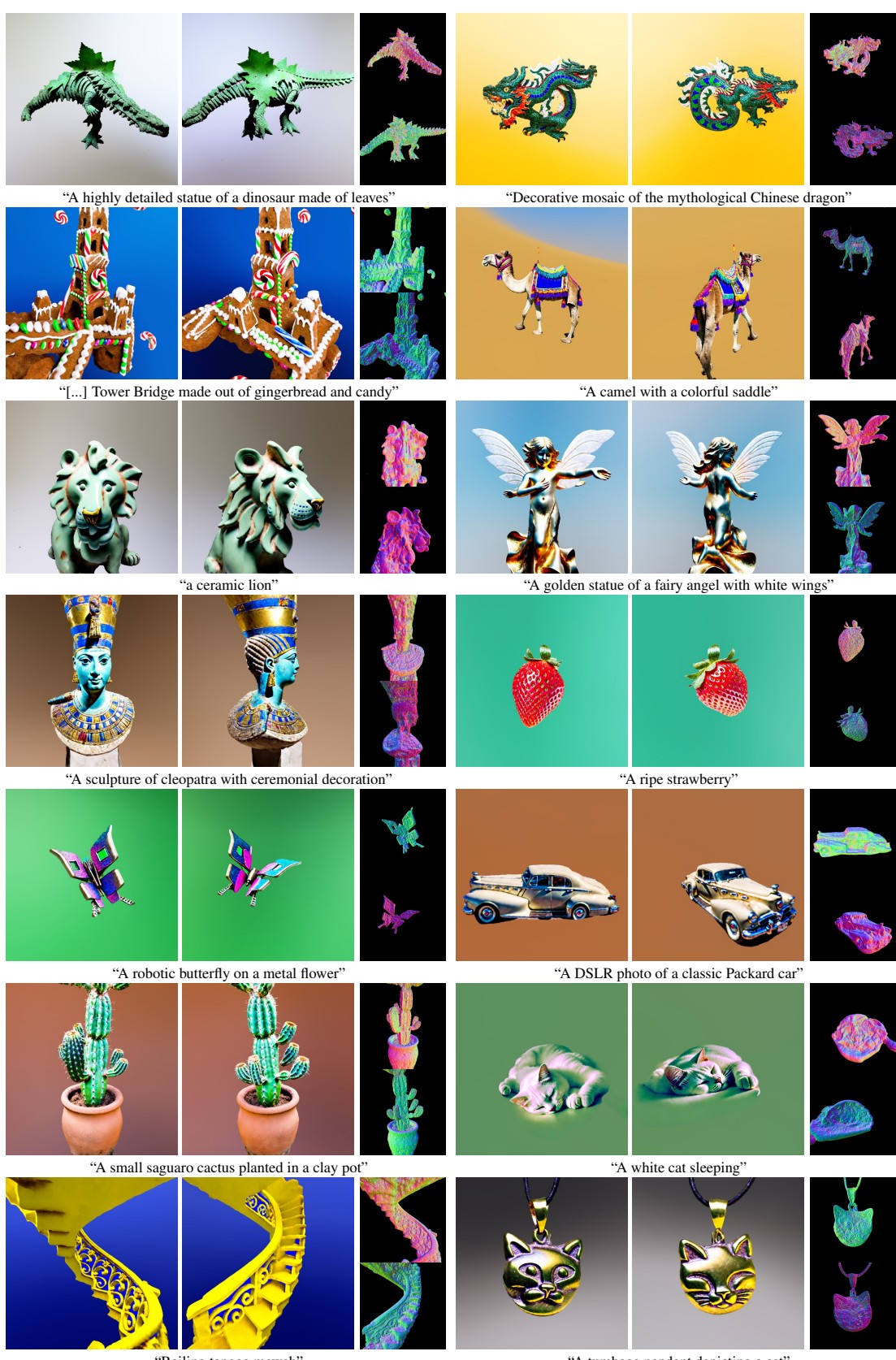

"A highly detailed statue of a dinosaur made of leaves"          "Decorative mosaic of the mythological Chinese dragon"

"[...] Tower Bridge made out of gingerbread and candy"          "A camel with a colorful saddle"

"a ceramic lion"          "A golden statue of a fairy angel with white wings"

"A sculpture of cleopatra with ceremonial decoration"          "A ripe strawberry"

"A robotic butterfly on a metal flower"          "A DSLR photo of a classic Packard car"

"A small saguaro cactus planted in a clay pot"          "A white cat sleeping"

"Railing tangga mewah"          "A tumbaga pendant depicting a cat"

Figure 15: NeRFs optimized with NFSD. [*] "A zoomed out DSLR photo of", [...]-"A wide angle zoomed out DSLR photo of zoomed out view of".

## A.8 MORE COMPARISONS WITH RELATED METHODS

As mentioned in the main paper, methods in different papers were implemented differently. For example, the diffusion model used in each of the methods if different, the NeRF implementation may be different, etc. Hence, we present results obtained by running the unified threestudio (Guo et al., 2023) framework implementation for all the methods. For all the methods we use Stable Diffusion 2.1-base and run them on a single GPU. As can be seen in Figure 16, for some of the methods the results look better with threestudio (e.g., DreamFusion), but for other methods the results may seem worse. For Magic3D (Lin et al., 2023) the gap may be caused by the diffusion model, while for Fantasia3D (Chen et al., 2023) it may be caused by the amount of resources. The results reported by the authors of Fantasia3D were obtained by optimizing on 8 GPUs while we use a single one. Overall, our method is comparable or better than all the methods, exhibiting high resolution and detailed features.

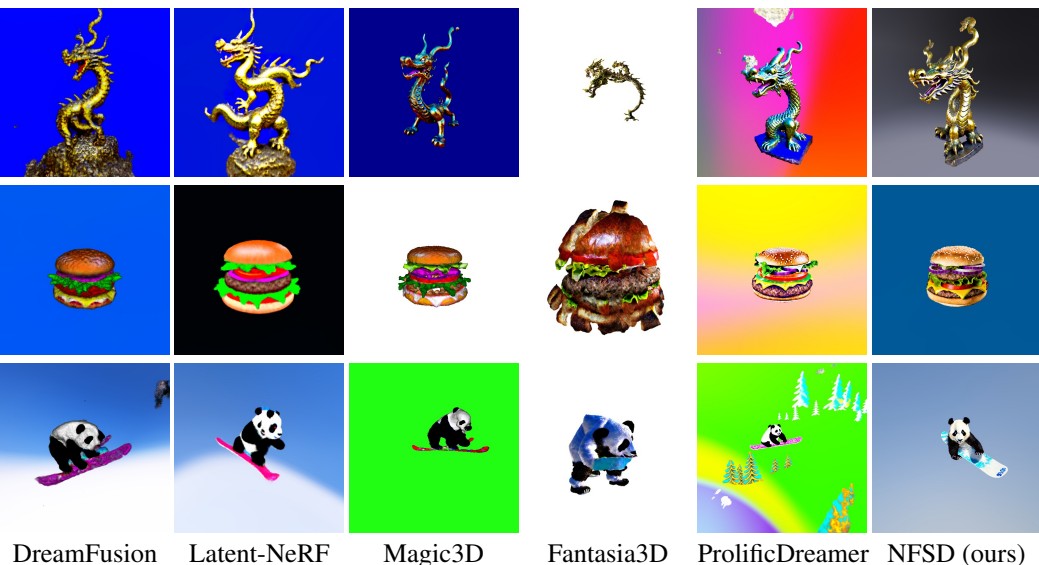

DreamFusion   Latent-NeRF   Magic3D   Fantasia3D   ProlificDreamer   NFSD (ours)

Figure 16: Comparison of NFSD with other methods using the threestudio (Guo et al., 2023) implementation.

Additionally, in Figures 17,18,19,20 and 21 we show additional comparisons with results obtained by other methods, as reported in their respective original papers.

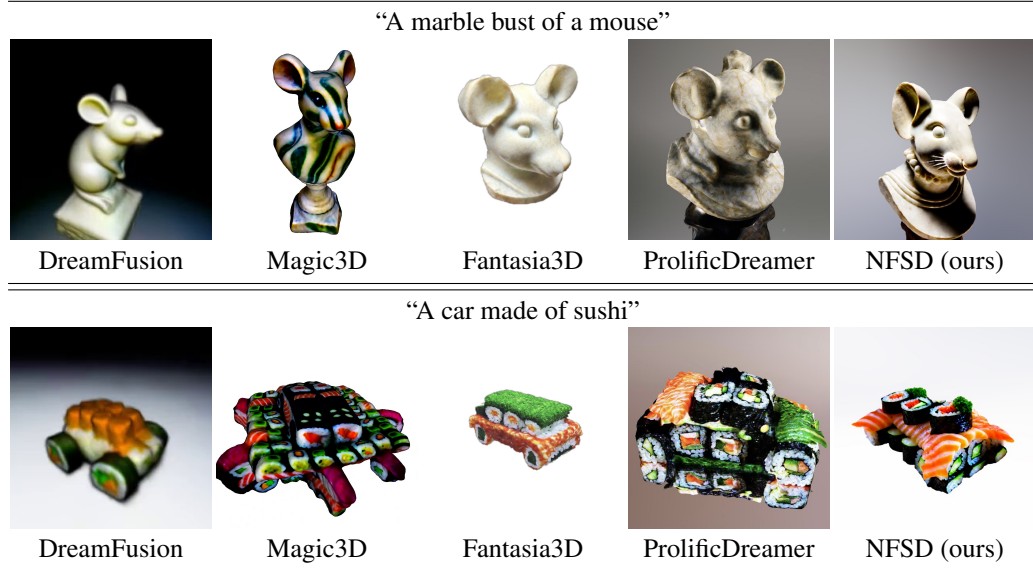

Figure 17: Comparison of NFSD with other methods using results obtained from the original papers.

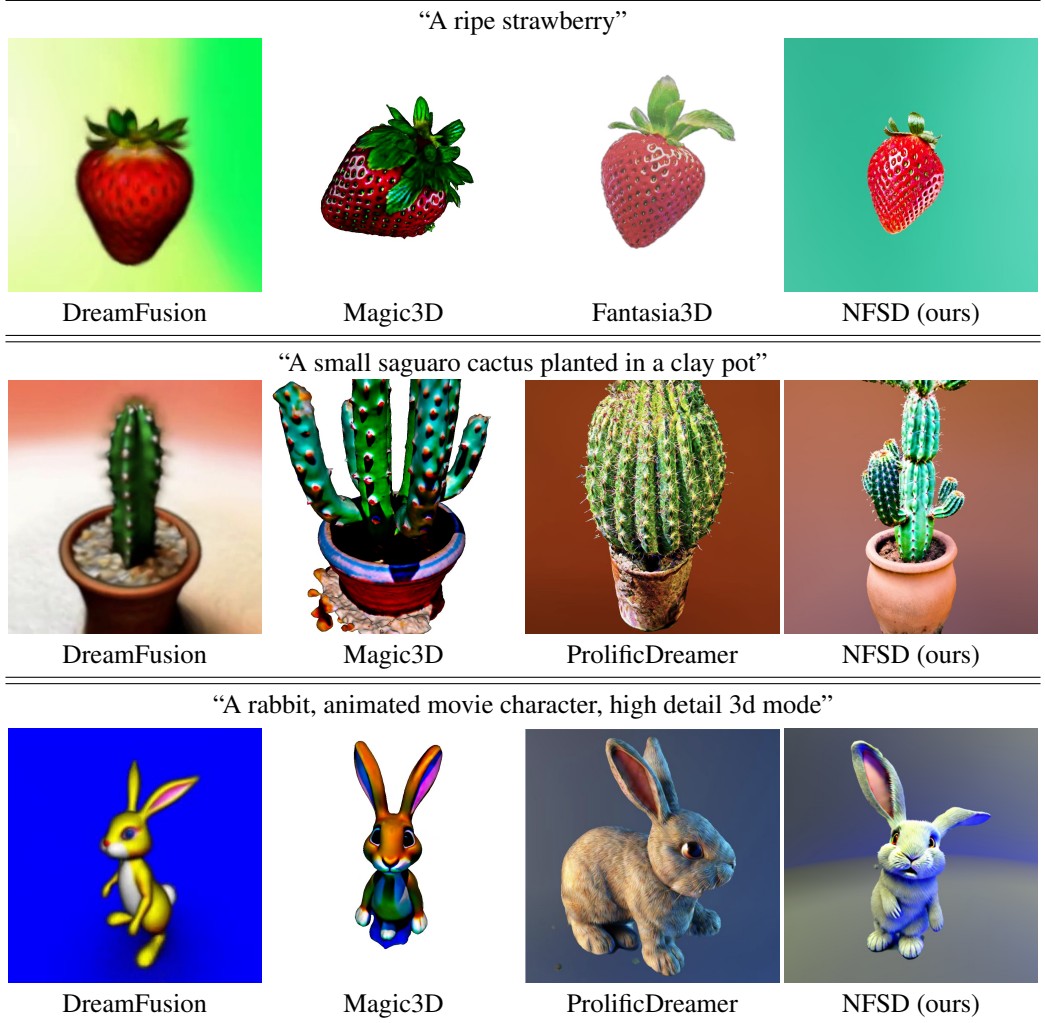

Figure 18: Comparison of NFSD with other methods using results obtained from the original papers.

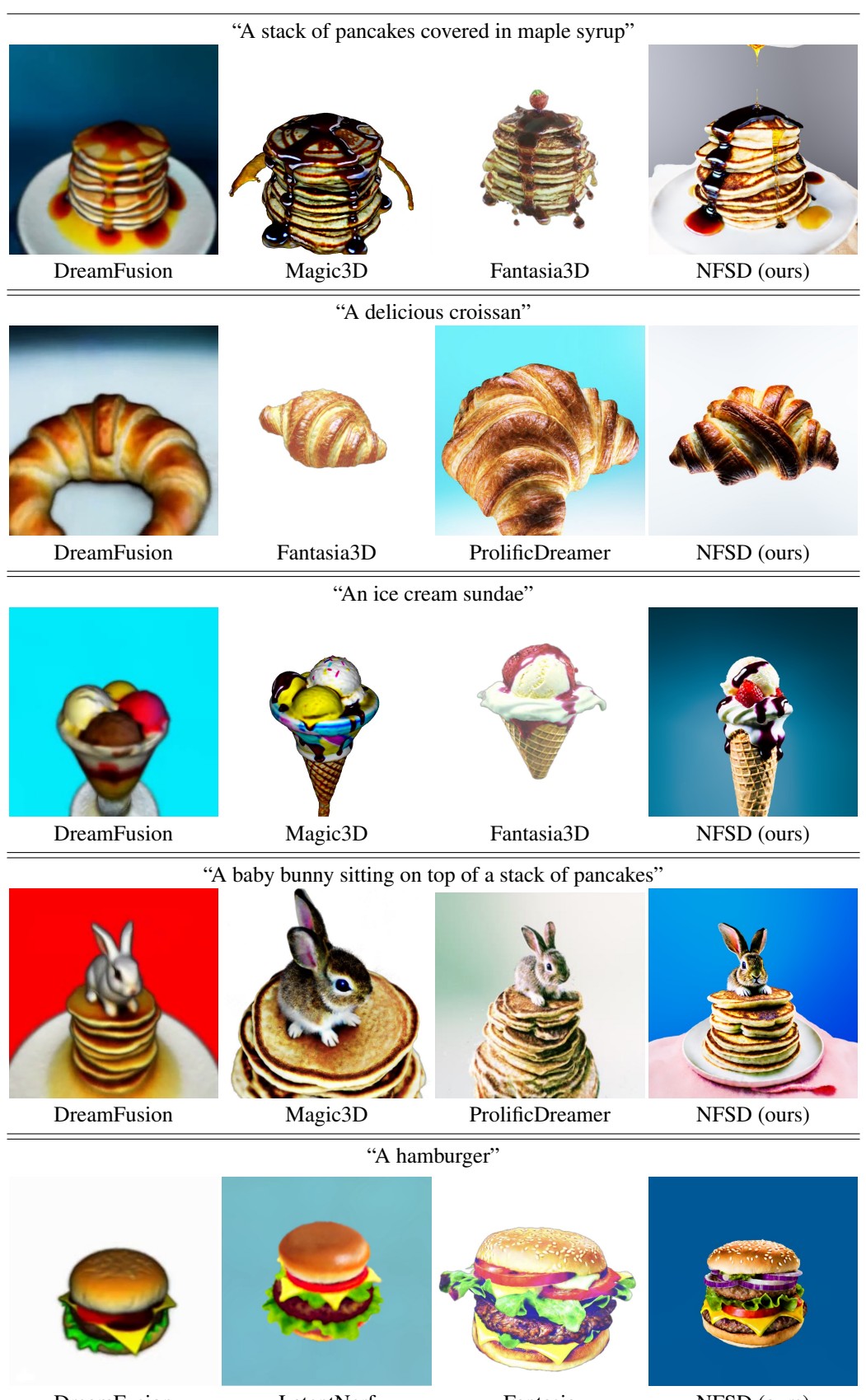

Figure 19: Comparison of NFSD with other methods using results obtained from the original papers.

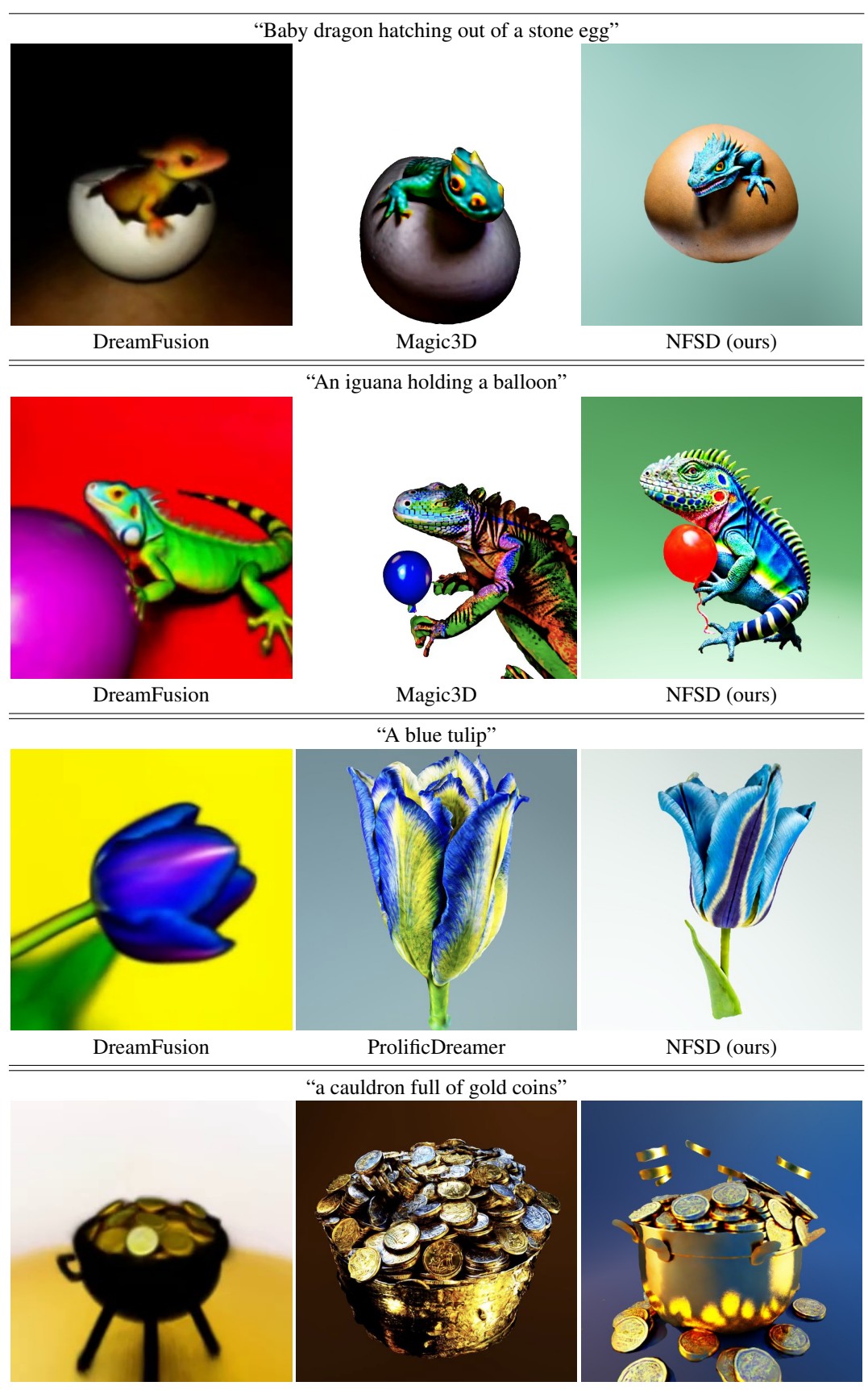

Figure 20: Comparison of NFSD with other methods using results obtained from the original papers.

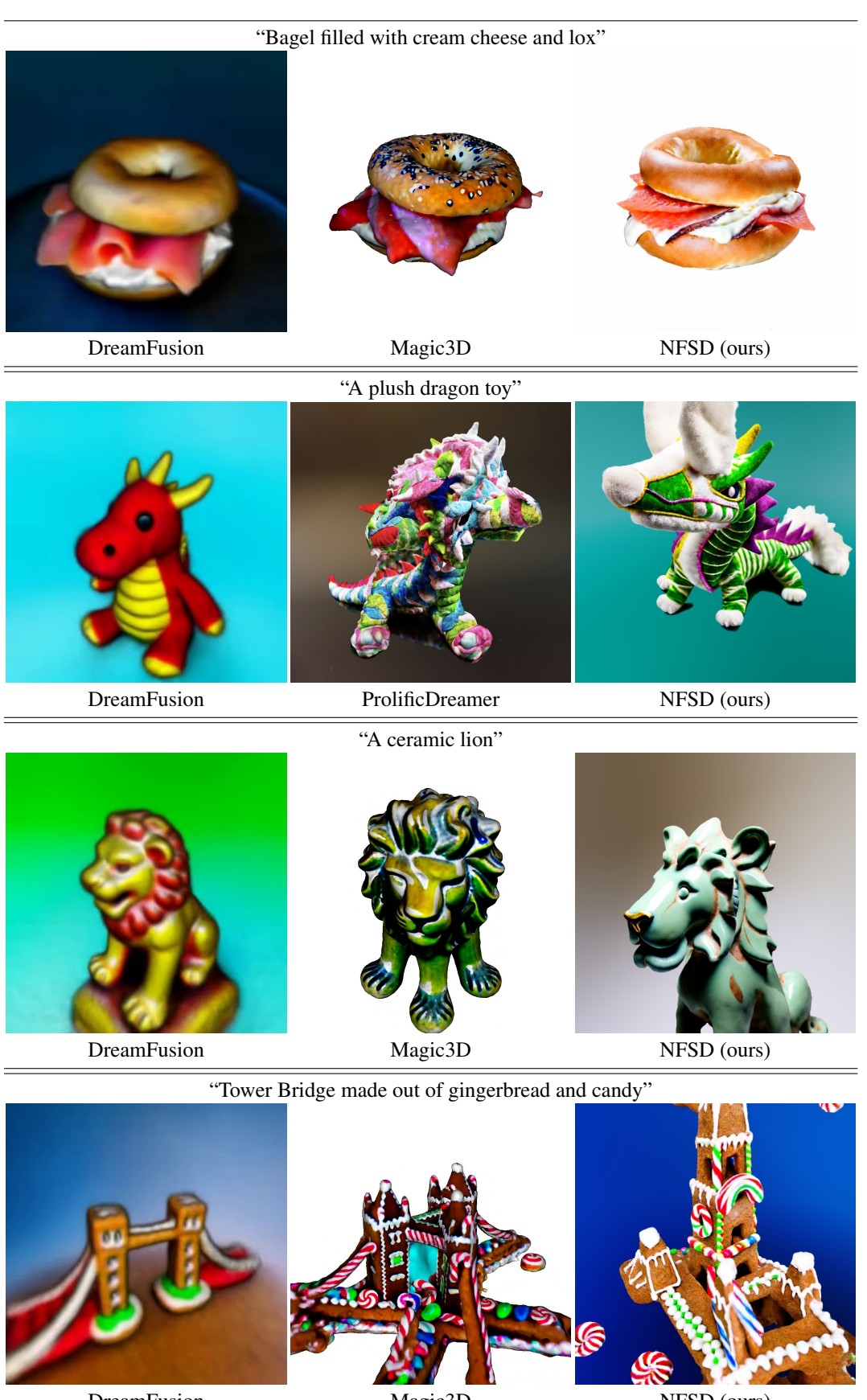

Figure 21: Comparison of NFSD with other methods using results obtained from the original papers.

