# OpenReview forum: "Noise-free Score Distillation"
_ICLR.cc/2024/Conference — ICLR 2024 poster_

### Official Review · Reviewer_EnYB · 2023-10-29

**Soundness:** 3 good
**Presentation:** 3 good
**Contribution:** 3 good
**Rating:** 6
**Confidence:** 3

**Summary:**

This work tackles the blurry results from Score Distillation Sampling (SDS) for text-to-3D generation. The score is decomposed into the condition, domain, and noise residual terms; the proposed method is designed to reduce the effect of the undesired noise component, by heuristically estimating the domain term with a negative prompt.

**Strengths:**

- Text-guided 3D generation by leveraging pretrained text-to-image models is a hot, timely topic. The authors propose to improve the famous SDS-based framework with a small modification.
- The proposed decomposition can be also used to understand existing works (DDS and VSD), which seems like a valuable contribution.
- The paper is well-written; the terms and derivations are clearly presented.

**Weaknesses:**

- Although a key value of this work seems to be the decomposition of the SDS loss, I have a few questions on designing the proposed ~NFSW~ NFSD method (<- apologize for this big typo at the initial review):
  - How did the authors separate the small and large timestep values based on t=200? Why not t=100, 300, or 400?
  - Is it valid to assume δ_{C=p_neg} ≈ −δ_{D}? Did the choice of the negative prompt affect the performance?
  - How about changing (6) into just using the second part of (6) for all the time steps (i.e., unconditional term - negative-prompt-induced term, for all the time steps)?
- I appreciate the effort for many visual results; however, the lack of any quantitative results concerns me a lot. Is it possible to include the comparison using CLIP R-Precision of Table 1 in the DreamFusion paper? Furthermore, leveraging the MS-COCO text-to-image benchmark with FID/IS/CLIP score metrics may be worth trying to justify the results of 2D image generation in Figure 7.

**Questions:**

Please refer to the Weaknesses section.

---

> ### Author Response · Authors · 2023-11-17
> **Reply to reviewer EnYB**
>
> Thank you for your time and constructive questions. We address them below:
>
>
> **How did the authors separate the small and large timestep values based on $t=200$? Why not $t=100, 300$, or $400$?**
>
> We added ablation studies in Appendix A.5, including an ablation on this threshold. As can be seen in the results presented in Figure 12, $t\_s=200$ provides the best tradeoff between fine details and artifacts.
>
> **Is it valid to assume $\delta\_{C=p\_\text{neg}} ≈ −\delta\_\text{D}$? Did the choice of the negative prompt affect the performance?**
>
> We added a discussion that motivates this choice in Appendix A.4. As explained there, there is a close relation between this assumption and the common technique of using negative prompts in ancestral sampling to improve image quality.
> The negative prompt was kept fixed during all experiments (both images and NeRFs). Our method is not sensitive to the exact choice of words in the negative prompt, and other combinations of negative words do not have much impact.
>
>
> **How about changing (6) into just using the second part of (6) for all the time steps (i.e., unconditional term - negative-prompt-induced term, for all the time steps)?**
>
> We added ablation studies in Appendix A.5. In the time threshold ablation presented in Figure 12 we included results for $t\_s=0$, in which the negative prompt term is applied for all timesteps.
>
> **Lack of any quantitative results concerns me a lot. Is it possible to include the comparison using CLIP R-Precision of Table 1 in the DreamFusion paper? Furthermore, leveraging the MS-COCO text-to-image benchmark with FID/IS/CLIP score metrics may be worth trying to justify the results of 2D image generation in Figure**
>
>
> As mentioned in our conclusion section, quantitatively comparing NeRFs generated from text prompt remains a challenge due to the absence of suitable metrics and benchmarks. Additionally, most works do not provide an official implementation which makes the quantitative evaluation even more tricky. Therefore, we chose to provide a large gallery of results (with prompts *taken* from the baseline works) to allow the reader to get an impression and to directly assess the quality of our results compared to other works.
> Yet, we add quantitative evaluation in Appendix A.6, measuring FID on generated images with NFSD and SDS-100, and presenting user study results, following Magic3D and ProlificDreamer.
> As can be seen in Table 1, our method achieves lower (better) FID compared to SDS-100, confirming the better quality achieved by our method.
> As can be seen in Table 2, our method also achieved the best results in both prompt alignment and image quality in the user study. We suspect that in the prompt alignment question responders were biased by the superior image quality of our results. Note that our results are aligned with the results shown in the user study in ProlificDreamer, in the sense that in both studies ProlificDreamer got better results than Fantasia3D, which, in turn, got better results than Magic3D.

---

> > ### Comment · Reviewer_EnYB · 2023-11-20
> >
> > Thanks for your effort and time on this rebuttal. I’ve reread the revised paper and other reviewers’ comments. I am generally satisfied with the rebuttal and appreciate the additional ablation studies and strengthened quantitative evaluation. I think I assigned my highest score in the initial review phase, and the following things would be good to address.
> >
> > - I appreciate the proposed score decomposition and the further discussion on the relationship to using negative prompts in Sect. A.4. It would be beneficial to add SDS-100+NP and NFSD+NP in Figure 7 and/or Table 1, since the effect of NP seems significant.
> >
> > - I appreciate Q1 raised by Reviewer eQLa and the subsequent discussion regarding diversity. I am looking forward to seeing the revision in the last paragraph of Sect. 3 (the issue of less diverse results).
> >
> > - This does not affect the score, but it would be great to see if the hyperparameters found for SD-2.1-base can be applicable to other SD versions with the same latent dimensions (e.g., SD-1.x using 64x64) or with enlarged dimensions (e.g., SD-2.1 using 96x96).

---

> > > ### Author Response · Authors · 2023-11-22
> > > **Reply to comment by Reviewer EnYB**
> > >
> > > Thank you for your feedback, below we address the points raised.
> > > * We have replaced Fig13 to include the examples from Fig7 with a negative prompt. Generating the required images for measuring FID in table 1 takes time and can be added to the next version of the paper.
> > > * In our revision, subsequent to the response by eQLa, we have modified the final paragraph of Section 3 concerning the association between CFG scale and diversity.
> > > * We've primarily concentrated on SD version 2.1.base. Thoroughly analyzing the effects of different diffusion models will take some time.  We intend to include this in the next version of our paper.

---

### Official Review · Reviewer_wypJ · 2023-11-01

**Soundness:** 4 excellent
**Presentation:** 4 excellent
**Contribution:** 3 good
**Rating:** 8
**Confidence:** 4

**Summary:**

This paper reexamined the Score Distillation Sampling and proposed Noise-Free Score Distillation. The details of the images generated using SDS are more blurred, due to the slightly different distribution between the images generated by the generator(x(\theta)) and the original image x. This paper found a decomposition to counteract this effect and the authors use this decomposition to explain why previous methods have improved SDS. Adequate experimental results also demonstrate the effectiveness of the methodology.

**Strengths:**

This paper proposed a decomposition method to solve the problem of ambiguous results caused by the different distribution of the images generated by the generator and the original images; and uses this decomposition method to explain why previous methods have improved SDS. The experimental results are intuitive.

**Weaknesses:**

I'm concerned about whether p_{neg} = “unrealistic, blurry, low quality, out of focus, ugly, low contrast, dull, dark, low-resolution, gloomy” is generalizable across situations and able to cancel out \delta_{N}. Would a better generator g(\theta) be able to achieve the same effect, or train a model to estimate the bias \delta_{N}?

**Questions:**

Please see the weaknesses.

---

> ### Author Response · Authors · 2023-11-17
> **Reply to reviewer wypJ**
>
> We thank the reviewer for their time and the concern they raised.  We have addressed it below.
>
> **I'm concerned about whether $p\_\text{neg}$ = “unrealistic, blurry, low quality, out of focus, ugly, low contrast, dull, dark, low-resolution, gloomy” is generalizable across situations and able to cancel out $\delta\_\text{N}$. Would a better generator $g(\theta)$ be able to achieve the same effect, or train a model to estimate the bias $\delta\_\text{N}$?**
>
> We evaluated our approach with two generators $g(\theta)$, one that renders NeRFs, and the other is defined by $g(\theta) = \theta$. We used the same $p\_\text{neg}$ in both cases and observed that it generalizes across them.
> It is worth mentioning that we observed that incorporating $\delta\_\text{D}$ for the directly optimized image was more significant compared with adding it to the optimized NeRF. This is related to the NeRF representation which naturally provides regularization on the rendered images. Therefore, it is true that our implementation for $\delta\_\text{D}$ affects differently on different generators $g$.
> As mentioned in Section 5, training a model that estimates $\delta\_\text{N}$ is exactly the method introduced in ProlificDreamer. We believe that presenting ProlificDreamer with this formulation helps to gain more intuition over their method, and see this as one of our contributions.

---

### Official Review · Reviewer_opTp · 2023-11-02

**Soundness:** 3 good
**Presentation:** 4 excellent
**Contribution:** 2 fair
**Rating:** 6
**Confidence:** 5

**Summary:**

This paper first revisits Score Distillation Sampling (SDS) and proposes to decompose the updates generated by SDS into three components: domain correction, noise estimation, and condition direction. Through this approach, the authors provide an explanation for why SDS accommodates a high Classifier-Free Guidance (CFG) coefficient and introduce Noise-Free Score Distillation (NFSD). NFSD re-estimates the unconditional score using the negative prompt trick. As a result, NFSD can employ a standard CFG weight to alleviate the over-smoothing/saturation problem and enhance the quality of text-guided image editing and 3D asset generation.

**Strengths:**

+ The paper is well structured and organized. The method introduced in this paper is intuitive and straightforward to implement. The motivations behind the approach are vividly conveyed through clear formulations and effective visualizations.

+ The decomposition of SDS is both novel and intriguing. It not only offers a compelling interpretation of the large CFG weight selection in DreamFusion but also offers valuable insights into DDS [1] and VSD [2].

+ The empirical results clearly demonstrate a significant enhancement in 3D generation through the simple modifications initiated by NFSD.

[1] Hertz et al., Delta Denoising Score, 2023

[2] Wang et al., ProlificDreamer: High-Fidelity and Diverse Text-to-3D Generation with Variational Score Distillation, 2023

**Weaknesses:**

- While the explanation is intuitively presented, it remains somewhat challenging to discern the fundamental distinction from the negative prompt trick.

- In Sec. 5, the paper asserts that NFSD is notably more efficient than VSD, despite sharing a similar working mechanism. Although this claim appears obvious, I would recommend providing quantitative evidence to substantiate this advantage when compared to other baseline methods. It is conceivable that dropping the noise term could even speed up the convergence of ancestral sampling by using fewer optimization steps.

- Further ablation studies are needed to validate the assertions put forth in this paper. In comparison to SDS, two terms have been omitted according to Eqs. 5 and 7: the noise prediction $\delta_N$ and the noise ground truth $\epsilon$. However, it remains unclear which of these terms plays the most pivotal role in improving the final results.

**Questions:**

1. Furthermore, it is not evident how steering the update of SDS could alter the optimization objective. Providing a more rigorous and formal argument would deepen the contribution of this work.

2. The authors introduce Eq. 6 to estimate $\delta_D$. Can the authors offer a rationale or justification for this approximation? Additionally, including visualizations that align with Fig. 3 would enhance the clarity and understanding of this proposal.

---

> ### Author Response · Authors · 2023-11-17
> **Reply to reviewer opTp**
>
> We thank the reviewer for their detailed comments and questions. We address them below:
>
> **It remains somewhat challenging to discern the fundamental distinction from the negative prompt trick.**
>
> Stemming from the presented score decomposition, our fundamental distinction is that $\delta\_\text{C}$ and $\delta\_\text{D}$ should be used to optimize the parameters $\theta$. While it is clear how to calculate $\delta\_\text{C}$, separating $\delta\_\text{D}$ from $\delta\_\text{N}$ remains a challenge.
> We added a discussion about negative prompts in Appendix A.4, motivating our definition of $\delta\_\text{D}$ by relating it to the common practice of negative prompts.
> Please note that we see the exact definition of $\delta\_\text{D}$ as an implementation detail for applying our main observation.
>
>
> **In Sec. 5, the paper asserts that NFSD is notably more efficient than VSD, despite sharing a similar working mechanism... I would recommend providing quantitative evidence. It is conceivable that dropping the noise term could even speed up the convergence of ancestral sampling by using fewer optimization steps.**
>
> For 2D-image generation, trained on a single RTX2080 for 1K iterations, on average:
>
> SDS - 2.9 [min]  NFSD - 4.3 [min] and Prolific - 8 [min]
>
> For NeRF generation, trained on a single A100 for 25K iterations, on average:
>
> SDS - 3.1 [hr], NFSD - 3.5 [hr] and Prolific - 4.5 [hr]
>
> Please note that we did not optimize our implementation.
> The noise term is undesirable for score distillation sampling. In contrast, for ancestral sampling, the score (comprising the noise term) is inherent to the step-by-step stochastic progression.
>
> **Further ablation studies are needed to validate the assertions put forth in this paper. In comparison to SDS, two terms have been omitted according to Eqs. 5 and 7: the noise prediction $\delta\_\text{N}$ and the noise ground truth  $\epsilon$. However, it remains unclear which of these terms plays the most pivotal role in improving the final results.**
>
> In practice, the difference between SDS and our NFSD is that for $t < t\_s$ we do not subtract the added noise $\epsilon$, where $t\_s=200$, and for $t >= t\_s$ we subtract $\epsilon(z\_t, p\_\text{neg}, t)$ (instead of subtracting $\epsilon$). We added ablation studies in Appendix A.5, where we ablate the following:
> * Subtracting $\epsilon(\mathbf{z}\_t, p\_\text{neg}, t)$ in all timesteps (instead of subtracting $\epsilon$ in SDS) ($t\_s=0$ in Figure 12)
> * Not subtracting $\epsilon$ from SDS loss in all timesteps ($t\_s=1000$ in Figure 12)
> * Change the threshold $t\_s$ (Figure 12)
> * Using only $\delta\_\text{C}$ as the loss (leftmost column in Figure 13)
>
>
> **It is not evident how steering the update of SDS could alter the optimization objective. Providing a more rigorous and formal argument would deepen the contribution of this work.**
>
> We developed NFSD with the perspective of score function distillation, aiming to distill the score of a diffusion model for optimizing the parameters of rendered images. We observed that the score distilled in SDS includes a noise component, resulting in an averaging effect, leading to a local optimum that is overly smooth and lacks detail. NFSD attempts to avoid this noise component, allowing the optimization process to converge to a better local optimum.
> The optimization objective for diffusion time $t < t_s$ is similar to SDS, while for $ t \geq t\_s$ it can be shown that $\nabla\_\theta \mathcal{L}\_{\textrm{NFSD}} =\nabla\_\theta \mathbb{E}\_{t}\left[w(t)\frac{\sigma\_t}{\alpha\_t}\textrm{KL}\left(q(\mathbf{z}\_t|g(\theta);y,t)\parallel \frac{1}{Z}\frac{p\_\phi(\mathbf{z}\_t;y,t)}{p\_\phi(\mathbf{z}\_t;y\_{neg},t)}\right)\right]$, where $Z$ is a normliazation factor.
> However, we find the optimized objective to lack intuitiveness and offer only little additional insight into NFSD. Therefore, we choose to focus on the score decomposition formulation.
>
>
> **The authors introduce Eq. 6 to estimate  $\delta\_\text{D}$ Can the authors offer a rationale or justification for this approximation? Additionally, including visualizations that align with Fig. 3 would enhance the clarity and understanding of this proposal.**
>
> Separating $\delta\_\text{D}$ from $\delta\_\text{N}$ is challenging. We found that for small enough timesteps (e.g., $t<200$) $\delta\_\text{N}$ is negligible and therefore $\epsilon(\mathbf{z}\_t, \varnothing, t) = \delta\_\text{N} + \delta\_\text{D}$ provides a good enough approximation for $\delta\_\text{D}$.
> For other timesteps, we approximate $\delta\_\text{D}$ by $-\delta\_{C=p\_\text{neg}}$. The discussion we added in Appendix A.4 motivates the use of a negative prompt to define $\delta\_\text{D}$.
> Additionally, to show the effect of our defined $\delta\_\text{D}$ in a similar manner to Figure 3, we added Figure 10  where we extract $\delta\_\text{D}$ from the definition in Eq. (6) (two rightmost images on the bottom row).

---

### Official Review · Reviewer_eQLa · 2023-11-06

**Soundness:** 3 good
**Presentation:** 3 good
**Contribution:** 3 good
**Rating:** 6
**Confidence:** 5

**Summary:**

This study proposes a simple yet effective method, Noise-Free Score Distillation (NFSD), to improve the conventional score distillation using a minimal modification. This study decomposes the score with classifier-free guidance (CFG)  into three terms, the condition, the domain, and the denoising components. Then, they remove the prediction error on unconditional samples between the estimated scores and and injected noises, since the score prediction error on unconditional samples is noisy. The domain score is estimated by a text prompt for a text-to-image model. The experimental results show that extremely high scale of CFG in score distillation is unnecessary, and NFSD can improve fine-grained details of generated images or neural fields.

**Strengths:**

S1. The proposed method, NFSD, is simple yet effective. In addition, the qualitative results support and demonstrate the effectiveness of NFSD.

S2. The paper is well-organized and easy to understand.

S3. The analogical decomposition of scores into three terms is interesting and makes sense.

**Weaknesses:**

W1. Despite the interestingness of score decomposition, the proposed method stems from numerous assumptions based on empirical findings without a principal approach.

W2. Thorough experiments to validate the effectiveness of NFSD are absent. Although the qualitative results show improved quality of text-to-NeRF than conventional SDS-based approaches, there is no ablation study and quantitative result.

W3. Some technical parts lack enough rationales. For example, estimating the domain score by negative text prompts lacks the rationales.

**Questions:**

Q1. Although the authors discuss the low diversity of NFSD, I wonder the detailed reason why the reduced CFG scale cannot produce diverse visual contents. In addition, can the authors provide the samples with different seeds and the same text prompts to show the diversity of generated contents?

Q2. In Figure 3, what is the diffusion timestep? In addition, I think that the authors should show the results of
$x_{\text{OOD}} + \delta_D + \delta_N^{\text{OOD}}$
, where $\delta_N^{\text{OOD}}$ is the denoising score of $x_\text{OOD}$, not $x_\text{ID}$. I also suggest clarifying the notation of $\delta_N$ and $\delta_D$ in Figure 3, since the two scores are from different samples.

Q3. Why do the prediction errors in Figure 4 (the second row) show a less-noisy map at t=1000? I think that the results are unintuitive, since they indicate that the score prediction at t=1 is difficult, while the score prediction at t=1000 is conducted almost perfectly except for the central region.

Q4. In Section 4, the authors claim that the magnitude of the noise to be removed is monotonically decreased in the backward process. I wonder how we can assume that the scale of the domain score is preserved? Is there any rationale that only $\delta_N$ decreases over the backward process, while $\delta_D$ preserves its scale?

Q5. How about the results of SDS, where its CFG adopts the same negative prompts as NFSD, described in Section 4?

Q6. The authors have discussed that ProlificDreamer’s LoRA adaptation has a similar role with NFSD to exclude the prediction error of the denoising term $\delta_N$. Then, can the LoRA of ProlificDreamer be replaced with NFSD, while variational particle optimization is used? It would be interesting to show the compatibility of NFSD with ProlificDreamer.

Q7. Since NFSD requires additional inference at each training iteration due to negative prompting, I think that comparing the results of NFSD with those of SDS in terms of the number of function evaluations (NFEs) of diffusion models.

Q8. In Section 4, how can we assume that the score prediction on text conditions is also composed of $\delta_D + \delta_N + \delta_C$, where $\delta_D + \delta_N$ is equal to the unconditional prediction? I think that it is a technical flaw, since Eq.(3) just implies $\epsilon_\phi (z_t ; t) - \epsilon_\phi (z_t  ; y=p_\text{neg}, t) = \delta_{C=p_\text{neg}}$. That is, $\delta_C$ is defined with both conditional and unconditional scores, not solely on the conditional score term.

Q9. How is the negative prompt to estimate the domain term defined? I wonder whether the negative prompt is universal regardless of the image renderer. In addition, it assumes that the domain score can be estimated by the text prompts. However, how can we say that the image is from out-of-distribution, when the image can be estimated by text prompts of text-to-image models?

Q10. Can be simply using $s \delta_C$ for the score distillation possible without $\delta_D$? That is, using $\delta_D$ is necessary?

---

> ### Author Response · Authors · 2023-11-17
> **Reply to reviewer eQLa**
>
> We thank the reviewer for their detailed comments and questions. We address them below:
>
> **W2. Thorough experiments to validate the effectiveness of NFSD are absent…There is no ablation study and quantitative result.**
>
> We added ablation studies in Appendix A.5 and quantitative evaluation in Appendix A.6.
>
> **Q1.Why the reduced CFG scale cannot produce diverse visual contents. In addition, can the authors provide the samples with different seeds and the same text prompts to show the diversity of generated contents?**
>
> Our approach actually sheds some light on the lack of diversity - it is not due to the high CFG. We hypothesize that the reason for the low diversity is rooted deeply in the process of score distillation. Unlike ancestral sampling, at each SDS iteration we add random noise to a rendered image, and we update the rendered image (or the generator parameters) independently of previous noises . Therefore, for a specific noised rendered image we follow some direction to optimize $p\_t(\mathbf{x}\_t)$. At the next step we again add (different) random noise and optimize $p\_t’(\mathbf{x}\_t’)$, which can yield a completely different direction. This results in an averaging effect (unlike ancestral sampling where each step corrects the previous as it is conditioned on it).
> Moreover, in our work, as in most score sampling methods, even the same diffusion time (i.e. similar noise level) can be used with different noises during the optimization process, yielding a different direction each time.
> We added a discussion alongside results obtained from different seeds in Appendix A.3.
>
>
> **Q2. In Figure 3, what is the diffusion timestep? ... show the results of $x\_{OOD}+\delta\_D + \delta\_N^{OOD}$ where $ֿ\delta\_\text{N}^{\text{OOD}}$  is the denoising score of $x\_{\text{OOD}}$ not $x\_{\text{ID}}$. I also suggest clarifying the notation of $\delta\_\text{D}$ and $\delta\_\text{N}$  in Figure 3, since the two scores are from different samples**
>
> The diffusion timestep used is $t=400$. Indeed, this should be specified in the paper.
> In Figure 3 we propose to visualize $\delta\_\text{N}$ and $\delta\_\text{D}$ by an engineered pair of $\mathbf{x}\_\text{ID}$ and $\mathbf{x}\_\text{OOD}$ images. We set $\delta\_\text{N}$ to be the denoising score of $\mathbf{x}\_\text{ID}$ as the image is in-domain and the prediction mostly just removes noise. We assume that $ֿ\delta\_\text{N}$ is shared between the in-domain and out-of-domain predictions, thus $\delta\_\text{D}$ is obtained as the difference between predictions of $\mathbf{x}\_\text{OOD}$ and $\mathbf{x}\_\text{ID}$. We explain this more clearly in the revised version.
> Regarding the term $\mathbf{x}\_\text{OOD}+\delta\_\text{D} + \delta\_\text{N}$, please note that this includes a noise direction term, thus the resulting image is noisy, as we show in Figure 10 in our revision.
>
>
> **Q3. Why do the prediction errors in Figure 4 (the second row) show a less-noisy map at t=1000?**
>
> Although the initial impression may appear counterintuitive, it's important to bear in mind that these are unnormalized disparities between the noise $\epsilon$ and the network's prediction $\epsilon\_{\phi}$—meaning, they haven't been adjusted in relation to the diffusion noise scheduler. Predicting the precise added noise to the image becomes notably more challenging when a small amount of noise is introduced. In contrast, at $t=1000$, the noised image predominantly consists of noise ($\epsilon \approx \mathbf{x}\_t$), simplifying the prediction process. as the image is heavily influenced by the noise itself.
>
>
> **Q4. How can we assume that the scale of the domain score is preserved? Is there any rationale that only  $\delta\_N$ decreases over the backward process, while $\delta\_\text{D}$ preserves its scale?**
>
> You are correct that there is no guarantee that $\delta\_\text{D}$ will preserve its scale. Nonetheless, our main point is that since the noise to be removed is monotonically decreasing in the backward process (according to the noise-scheduler), the noise prediction $\delta\_\text{N}$ part of the score (corresponding to $\epsilon\_{\phi}$) is becoming increasingly negligible as diffusion time becomes smaller. In contrast, the magnitude of the $\delta\_D$ component depends both on the diffusion time and the current stage of the optimization process. Thank you for this important comment, and we have revised our paper accordingly.
>
>
> **Q5. How about the results of SDS, where its CFG adopts the same negative prompts as NFSD, described in Section 4?**
>
> We added a discussion about negative prompts in Appendix A.4. As we explain there, using a negative prompt in the CFG term mimics the effect of $\delta\_\text{D}$.
> Additionally, in Figure 13 we perform an experiment where we add a negative prompt to the CFG term of both SDS-100 and NFSD losses. As can be seen, the negative prompt improves the results of both SDS and NFSD.
>
> **(Continued in next reply)**

---

> ### Author Response · Authors · 2023-11-17
> **Reply to reviewer eQLa - cont.**
>
> **Q6. Can the LoRA of ProlificDreamer be replaced with NFSD, while variational particle optimization is used?**
>
> The variational particle optimization proposed in ProlificDreamer is entangled with the LoRA variational model, as it is designed to model possible 3D representations (rather than one).
> Hence, applying NFSD for multiple particles can not be directly transferred from ProlificDreamer to our method.
> A possible approach for multiple particles was proposed by [Kim et al](https://arxiv.org/abs/2307.04787). This is a general multi-particle approach for SDS-based methods, which can be combined with NFSD, but is out-of-the scope of our work.
>
>
> **Q7. Comparing the results of NFSD with those of SDS in terms of the number of function evaluations (NFEs) of diffusion models.**
>
> To the best of our knowledge, there isn't a method available for automatically determining the optimal number of iterations for SDS-based methods.
> In all our experiments, we use the same fixed number of iterations both for SDS and NFSD, resulting in an equal count of parameter updates for both methodologies.
> The term "NFEs" of diffusion models might be misleading here, given that the outcome of the diffusion model functions merely as a guiding factor without involving subsequent backpropagation through it. Theoretically, NFSD necessitates an additional evaluation of the negative prompt (in the worst-case scenario, as the negative prompt isn't utilized for diffusion times below $t\_s = 200$). Consequently, NFEs(NFSD) equate to 1.5 times NFEs(SDS).
> Increasing the iteration count did not yield improvements in the SDS results; On the contrary, it tended to introduce artifacts.
>
>
> **Q8. In Section 4, how can we assume that the score prediction on text conditions is also composed of  $\delta\_\text{D} + \delta\_\text{N} + \delta\_\text{C}$, where $\delta\_\text{D} + \delta\_\text{N}$ is equal to the unconditional prediction?**
>
> Thank you for the clarification. Indeed, $\delta\_\text{C}$ is defined as $\epsilon\_\phi(\mathbf{z}\_t;y,t) - \epsilon\_\phi(\mathbf{z}\_t;\varnothing,t)$. Plugging $y=p\_{\text{neg}}$ into the definition directly gives $\epsilon\_\phi(\mathbf{z}\_t;\varnothing,t) - \epsilon\_\phi(\mathbf{z}\_t;y=p\_\text{neg},t) = -\delta\_{c=p\_\text{neg}}$. Therefore, applying the score decomposition on the terms before Equation (6) is unnecessary and we fixed it in the revision.
>
>
> **Q9. How is the negative prompt to estimate the domain term defined?
> How can we say that the image is from out-of-distribution, when the image can be estimated by text prompts of text-to-image models?**
>
> We use the same prompt for all experiments (3D and 2D). The prompt is simply a collection of synonym words for “low quality” image.
> Given a specific prompt, e.g. “a photo of a panda”, the score of the diffusion model will be in the direction that maximizes the likelihood of the sample given the prompt, i.e. it will point to a dense region of the conditioned distribution. The term out-of-distribution refers to this conditioned distribution. While low quality images with noticeable artifacts can be generated by a diffusion model (simply input the negative prompt), under a “normal” condition e.g. “a photo of a panda”, this is highly unlikely and thus we term it as out-of-distribution.
>
>
> **Q10. Can be simply using $s\delta\_\text{C}$   for the score distillation possible without $\delta\_\text{D}$? That is, is using  $\delta\_\text{D}$  is necessary?**
>
> We added ablation studies in Appendix A.5, including an experiment for using $s\delta\_\text{C}$ only for 2D-latent in Figure 13. While the generated image is consistent with the prompt it contains artifacts, indicating that $\delta\_\text{D}$ term is necessary for obtaining high-quality results.
> We do stress that the effect of $\delta\_\text{D}$ can be achieved by other means of regularization, not only the diffusion model. For example, using additional losses for smoothness regularization can be used. Specifically in NeRF optimization, the parameter space and other losses (e.g. sprasity and opacity losses) can be used to regularize the results, serving as $\delta\_\text{D}$.

---

> ### Comment · Reviewer_eQLa · 2023-11-20
>
> I carefully read the authors' responses and the revised version of the paper including the Appendix.
>
> Regarding to Q1, I encourage the authors to further revise the paper. In the last paragraph in Section 3, “However, high CFG coefficients are known to yield less realistic and less diverse results, as demonstrated in Figure 5, typically leading to over-saturated images and NeRFs.” can lead to misunderstanding of authors, high CFG coefficients are the root cause of low diversity even in SDS.
>
> I have recognized the contributions of this paper, but the major concerns to reject this paper were some unclear explanations under assumptions and the lack of ablation study.
> However, the authors' responses clearly resolve most of my concerns, so I readily revise my final score to give an opinion to accept this paper.

---

> > ### Author Response · Authors · 2023-11-22
> > **Reply to comment by Reviewer eQLa**
> >
> > We appreciate the reviewer's careful review of our approach and the revision of the rating.
> > We have revised the last paragraph of Section 3, as suggested, to clarify that the use of a high CFG scale is not the root cause for lack of diversity in SDS.

---

### Author Response · Authors · 2023-11-17
**General response to reviewers**

We would like to thank all reviewers for their detailed reviews and constructive remarks.

We are excited to see that the reviewers found the score decomposition to be intriguing and interesting (eQLa, opTp) and appreciated the insights it offers into DDS and ProlificDreamer (opTp, wypJ, EnYB). The score decomposition is built on intuitive observations rather than rigorous derivations, and is mainly supported by experimental results (wypJ) and effective visualizations (opTp).
The reviewers found the paper to be well organized and clear (eQLa, opTp, EnYB), with a simple method (eQLa, opTp, EnYB) that demonstrates a significant enhancement in 3D generation (opTp).
The profound questions raised by the reviewers are addressed in our individual responses to each of the reviewers, and we believe that they helped to further support the method and improve the quality of the paper.
In addition to some clarifications we made in the revision, following questions raised by the reviewers we added a figure in Appendix A.1, and added Appendices A.3, A.4, A.5, A.6.

---

### Meta-Review · Area_Chair_ptbg · 2023-12-18

**Metareview:**

The paper provides interesting empirical results on how a simple modification to the Score Distillation Sampling framework can improve performance. The reviewers all recommend accept with several useful suggestions provided for improvement. We recommend the authors to incorporate all the feedback in future versions. It would be especially good to describe the assumptions, and intuitions carefully.

**Justification For Why Not Higher Score:**

The paper makes a neat empirical observation but lacks a solid conceptual/theoretical framing, in its current presentation form.

**Justification For Why Not Lower Score:**

The findings are interesting, empirical observations and conclusions seem sound, all reviewers recommend acceptance

---

### Decision · Program_Chairs · 2024-01-16

Accept (poster)